# TOOLBRIDGE: AN OPEN-SOURCE DATASET TO EQUIP LLMS WITH EXTERNAL TOOL CAPABILITIES

## ABSTRACT

Through the integration of external tools, large language models (LLMs) such as GPT-4o and Llama 3.1 significantly expand their functional capabilities, evolving from elementary conversational agents to general-purpose assistants. We contend that the primary drivers of these advancements are the quality and diversity of the training data. However, the existing LLMs with external tool integration provide only limited transparency regarding their datasets and data collection approaches, which has led to the initiation of this study. Specifically, in this work, we endeavor to present a detailed exposition of the methodology for constructing datasets that facilitate LLMs in effectively learning how to utilize external tools and make this process available to the public through the introduction of ToolBridge. ToolBridge proposes to leverage a collection of general open-access datasets as its raw dataset pool and incorporates a series of strategies to identify the appropriate data entries for external tool API insertions. By supervised fine-tuning (SFT) on these curated data entries, LLMs can invoke external tools in appropriate contexts to boost their predictive accuracy, particularly for essential functions including factual retrieval, data processing and numerical computation. Our experiments meticulously isolate model architectures and training configurations, zeroing in exclusively on the role of data. The experimental results indicate that LLMs trained on ToolBridge exhibit consistent performance gains on both standard benchmarks and custom evaluation datasets. All associated code and data will be released as open source, promoting transparency and facilitating the broader community to explore methodologies for equipping LLMs with external tools capabilities.

## 1 INTRODUCTION

Large language models (LLMs) have revolutionized natural language processing, excelling in tasks including question answering, summarization, and text generation Jiang et al. (2023); Achiam et al. (2023); Dubey et al. (2024); Gunter et al. (2024); Team et al. (2024). Despite the impressive achievements of LLMs, they persistently underperform in fundamental areas, such as arithmetic and factual lookup, where external tools can effectively provide solutions Schick et al. (2024).

Consequently, there has been a rise in research efforts committed to equipping LLMs with the ability to utilize external tools. These efforts can be broadly categorized into two distinct areas: (1) function calling, such as Toolllm Qin et al. (2023) and API-Bank Li et al. (2023), which offer curated datasets and standardized evaluation protocols to systematically assess and compare the tool-use capabilities of LLMs, focusing on enabling LLMs to function as agents that call APIs without requiring explicit training on their usage. (2) tool former, exemplified by GPT-4o and Toolformer Schick et al. (2024), which seek to empower LLMs with the capability to autonomously invoke and utilize external tools during inference, emphasizing advanced reasoning by enabling LLMs to discern which external tool to call, the appropriate timing for the call, and the systematic parsing and integration of the execution results into the generated text. This study primarily concentrates on the latter area.

Within the latter context, Llama 3.1 Dubey et al. (2024) is trained to utilize Brave Search, Wolfram Alpha API and Python interpreter by incorporating synthetic data combined with human-annotated data. Toolformer Schick et al. (2024) adopts a self-supervised training strategy to equip LLMs with the capability to leverage diverse tools effectively. However, these existing works tend to either only open-source their trained models and code or restrict access to the models through usage interfaces

**User:** "Calculate the answer to: 5^2."

**Assistant**: "The answer to 5^2 is `<python>answer = 5**2\nprint(answer)</python><result>25</result>` 25."

**User:** "Standardize the following date to the ISO 8601 format.\nJuly 26th, 2021."

**Assistant**: "The standardized date in ISO 8601 format is: `<python>from datetime import datetime\ndate_str = 'July 26th, 2021'\ndate_obj = datetime.strptime(date_str, '%B %dth, %Y')\niso_date = date_obj.date().isoformat()\nprint(iso_date)</python><result>2021-07-26</result>` 2021-07-26."

**User:** "Question: What's the answer to that question: what currency do the ukraine use?\nAnswer:"

**Assistant**: "The currency used in Ukraine is the `<python>import requests\n\nresponse = requests.get('https://restcountries.com/v3.1/name/Ukraine')\ncurrency = response.json()[0]['currencies']\ncurrency_code = list(currency.keys())[0]\ncurrency_name = currency[currency_code]['name']\nprint(currency_name)</python><result>Ukrainian hryvnia</result>` Ukrainian hryvnia (UAH)."

**User:** "Solve 5*z - 180 = -165 for z. What is the answer?"

**Assistant**: "To solve for z in the equation 5*z - 180 = -165, we need to isolate z on one side of the equation. \n\nFirst, we can add 180 to both sides of the equation to get rid of the constant term on the left side: \n\n5*z - 180 + 180 = -165 + 180 \n\nSimplifying the left side, we get: \n\n5*z = 15 \n\nNext, we can divide both sides of the equation by 5 to isolate z: \n\n5*z/5 = 15/5 \n\nSimplifying, we get: `<python>from sympy import symbols, Eq, solve\nz = symbols('z')\nequation = Eq(5*z - 180, -165)\nsolution = solve(equation, z)\nprint(solution[0])</python><result>3</result>`\n\nz = 3 \n\nTherefore, the solution to the equation 5*z - 180 = -165 is z = 3."

Figure 1: **Examples of data entries from ToolBridge.** The proposed pipeline is capable of generating a large volume of data entries to facilitate model learning to utilize external tools.

Achiam et al. (2023); Dubey et al. (2024); Gunter et al. (2024). The data underlying the training of these models is commonly withheld from public disclosure, enveloped in secrecy.

The widespread availability of pre-trained models stands in stark contrast to the limited transparency surrounding their training datasets, which impedes the community's progress in advancing discoveries in this area. This emphasizes the critical need to explore ways to create high-quality, publicly available datasets for training LLMs to utilize external tools to facilitate their reasoning process.

In response to the challenges discussed, this paper proposes a pipeline for the large-scale creation of datasets tailored to equip LLMs with the capability to effectively utilize external tools. Specifically, we begin by aggregating a substantial collection of open-source datasets used for LLMs supervised fine-tuning (SFT) from the community, which circumvents proprietary concerns including copyright issues. Upon establishing the dataset pool, we propose a systematic strategy to assist in identification of valuable data entries and convert them into a standardized format. Finally, consistency validation is conducted to further boost the quality of the converted data entries and ToolBridge is constructed. As demonstrated in Figure 1, we showcase the data entries from ToolBridge, which function to guide LLMs in understanding how to incorporate external tools in appropriate contexts, thereby improving the accuracy and reliability of their outputs. For instance, the third case in Figure 1 serves to instruct LLMs in leveraging the *requests* module to gather factual information from web sources. Moreover, LLMs can employ the fourth scenario to understand how *sympy* library can be applied to solve linear equation of one variable.

In summary, the contributions of this paper are as follows,

- We propose a pipeline capable of producing large volumes of entries for training LLMs to incorporate various external tools. The collection of over 178K yielded data entries, named ToolBridge, will be open-sourced to the community, marking a significant advancement in the transparency and accessibility of the data for training LLMs to leverage external tools.

- Our experimental results demonstrates that LLMs supervised fine-tuned on ToolBridge can achieve consistent performance improvements on several standard benchmarks.

- We also propose to curate a set of data entries to examine the abilities of LLMs in numerical calculation, data processing and factual retrieval before and after supervised fine-tuning on ToolBridge. Experimental outcomes reaffirm the effectiveness of ToolBridge

This is, to our knowledge, the first work in the domain of enabling LLMs to learn to utilize external tools that open-sources the training data. We anticipate that ToolBridge will facilitate the community

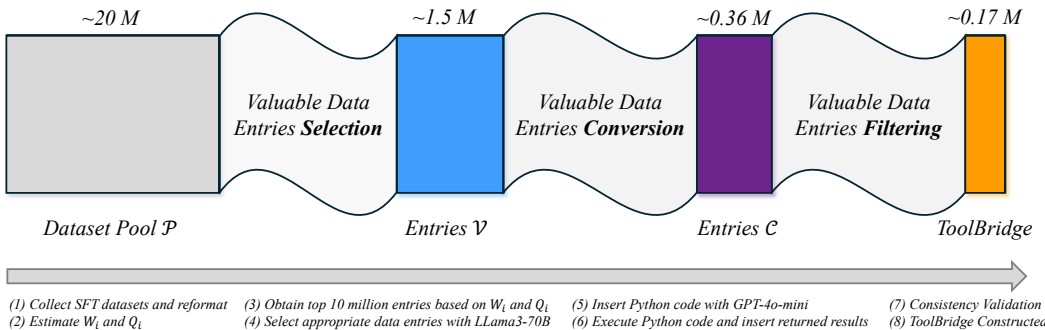

Figure 2: **Overview of the construction pipeline for ToolBridge.** The construction of ToolBridge follows three main steps: identifying valuable data entries in the dataset pool, converting these data entries through the integration of external tool calls and finally conducting a thorough cleanup of the converted data entries by consistency validation.

in further investigating the ability of LLMs to use external tools, thereby advancing LLMs from basic conversational models to versatile general-purpose assistants.

## 2 RELATED WORK

**Tool Use for LLMs.** Enabling LLMs to use external tools like search engines and code interpreters significantly broadens the range of tasks LLMs can address and strengthens their predictive accuracy. The methodologies for equipping LLMs with the capability to employ external tools can be broadly categorized into two paradigms, *i.e.*, function calling and tool former. In particular, function calling emphasizes allowing LLMs to act as agents that invoke APIs using predefined functions and prompts without necessitating explicit training on their usage Li et al. (2023); Shen et al. (2024). Conversely, tool former prioritize empowering LLMs with the ability to autonomously identify appropriate tools, determine the optimal invocation timings, and incorporate the outputs into their reasoning processes. For example, Komeili (2021) proposed to enable LLMs to adopt a search engine by learning to yield an internet search query based on the context, and then condition its generated response on the search results. Cobbe et al. (2021) facilitated LLMs' utilization of a calculator during inference by training the models with calculation annotations injected into the datasets. Thoppilan et al. (2022) proposed to assist LLMs in invoking external tools from a toolset, comprising an information retrieval system, a calculator and a translator, by training it to produce a special string *TS*. Gao et al. (2023) suggested adopting LLMs to interpret natural language problems and yield programs as intermediate reasoning, while delegating the solution process to a runtime environment like a Python interpreter. Toolformer Schick et al. (2024) allowed LLMs to learn how to adopt the external tools through a self-supervised learning approach. Of late, the works like GPT-4o, Llama 3.1 Dubey et al. (2024) and Apple LLMs further strengthened LLMs' ability to leverage external tools through improvements in training data, model architectures, *etc*. This study falls into the latter category, namely tool former.

Although previous research in the domain of tool former are highly praiseworthy, they seldom make the data required for training their models publicly available, which is crucial for the community to advance research and build upon their contributions. This paper presents a pipeline aimed at yielding data entries for training models in external tool utilization, along with open-sourcing all data entries produced using this methodology This open access facilitates the development of more effective and efficient algorithms for the next generation of LLMs integrated with external tool functionalities.

**Training Datasets for Tool Use.** Previous datasets designed to train LLMs to utilize external tools primarily fall within the function calling paradigm. For instance, Qin et al. (2023) collected a high-quality instruction-tuning dataset ToolBench, which is constructed automatically adopting ChatGPT. Li et al. (2023) introduced API-Bank, which encompasses 1,888 tool-use dialogues from 2,138 APIs spanning 1,000 distinct domains. However, to the best of our knowledge, there is a notable absence of research efforts that have open-sourced training datasets within the tool former domain.

To address this significant gap, this paper presents ToolBridge - a dataset of more than 178,000 data entries to support LLMs in effectively learning to utilize external tools within tool former paradigm.

## 3 TOOLBRIDGE

Previous LLMs like GPT-4o and Llama 3.1 only provide limited information on how they curate the data entries to empower themselves to employ external tools. To address the lack of transparency in training data, we propose a generic pipeline for constructing large-scale datasets from public sources to enable LLMs to use external tools. As indicated in Figure 1, the whole pipeline follows three main steps: valuable data entries **selection**, **conversion** and **filtering**.

| Source | # of Entries | Source | # of Entries |
|---|---|---|---|
| School Math 0.25M | 248,481 | LIMA | 1,330 |
| code_instructions_120k_alpaca | 121,959 | TigerBot | 1,199,030 |
| Platypus | 24,926 | TSI-v0 | 5,607,620 |
| ShareGPT90K | 90,665 | LaMini-Instruction | 2,585,615 |
| WizardLM_Orca | 54,974 | Bactrian-X | 67,017 |
| WizardLM_evol_instruct_70k | 70,000 | Baize | 210,311 |
| tiny-codes | 1,632,309 | COIG | 178,246 |
| WizardLM evolve_instruct V2 | 143,000 | MOSS SFT | 1,074,551 |
| No Robots | 10,000 | AlpacaDataCleaned | 51,760 |
| ign_clean_instruct_dataset_500k | 508,620 | GPT-4all | 808,812 |
| GPT-4-LLM | 113,003 | Alpaca | 52,002 |
| ChatAlpaca | 20,000 | self-instruct | 82,439 |
| OpenOrca | 4,233,923 | | |

Table 1: The composition of our dataset pool $\mathcal{P}$ to construct ToolBridge.

### 3.1 DATASET POOL CONSTRUCTION

Our work starts with a review of the data accessible in the community for the purpose of supervised fine-tuning (SFT). Table 1 summarizes the results. Owing to the diverse range of teams contributing the SFT datasets, there is significant heterogeneity in their formats, which introduces difficulties on effective model training. Hence, we first reformat all candidate datasets into a standardized ChatML format for further processing:

```
data entry = [{"role": "user", "content": "..."},
              {"role": "assistant", "content": "..."},
              ...].
```

After reformatting all datasets, we construct the dataset pool as $\mathcal{P} = \{(\mathcal{D}_i, W_i, Q_i) \mid i \geq 0\}$, where $\mathcal{D}_i$ denotes one candidate dataset, $W_i$ measures the proportion of valuable entries for each dataset, and $Q_i$ serves as a metric for assessing the quality of each dataset.

Practically, to obtain $W_i$, we first perform random sampling on $\mathcal{D}_i$ to produce a subset $\mathcal{S}_i$, containing 1% data entries of $\mathcal{D}_i$. Llama3-70B is then applied to judge the appropriateness of each entry in $\mathcal{S}_i$ for external tool invocation to enhance reasoning, where the prompt employed is shown in Appendix A.1. In generally, if an entry is deemed suitable for invoking external tools to help LLMs' reasoning process, we label it as a valuable entry. At last, we determine $W_i$ as the ratio between the number of valuable data entries and the total number of data entries in $\mathcal{S}_i$.

Additionally, we observe that some candidate datasets within the dataset pool $\mathcal{P}$ are partially sourced from the Internet via web scraping, resulting in the inclusion of certain meaningless HTML tags and other irrelevant content. Thus, we incorporate $Q_i$ as an additional metric to evaluate the data quality of each candidate dataset. To compute $Q_i$, we randomly sample $N$ data entries from $\mathcal{D}_i$ and conduct a manual review to identify any presence of irrelevant characters or content, where we configure $N$ as 100 by default. $Q_i$ is then derived as the fraction of data entries devoid of irrelevant contents over the total number of sampled entries $N$.

### 3.2 VALUABLE DATA ENTRIES SELECTION

Upon constructing the dataset pool $\mathcal{P}$, due to the large scale of candidate data entries, we propose to select 10 million data entries from $\mathcal{P}$ for further processing tailored to the dataset attributes $W_i$ and

$Q_i$. Particularly, we first arrange $\mathcal{D}_i$ in descending order guided by the value of $Q_i \times W_i$. Then, the data entries are selected from the top-ranked datasets sequentially until the overall volume amounts to 10 million data entries.

Subsequently, Llama3-70B with the prompt detailed in Appendix A.1 is applied to ascertain whether each entry within the 10 million samples is appropriate for LLMs to enhance reasoning via utilizing external tools. And we represent the collection of these appropriate data entries with $\mathcal{V}$, namely, the valuable data entries. In Table 2, we present the distribution of $\mathcal{V}$ across the respective datasets from which they are derived. It is observed that leveraging the capabilities of Llama3-70B, we can refine the 10 million data entries down to 1,527,153 valuable entries.

| Source | # of Entries | Source | # of Entries |
|---|---|---|---|
| School Math 0.25M | 205,996 | ChatAlpaca | 2,643 |
| Platypus | 7,776 | ShareGPT90K | 24,348 |
| WizardLM_Orca | 8,659 | WizardLM_evol_instruct_70k | 28,293 |
| WizardLM evolve_instruct V2 | 5,399 | MOSS SFT | 136,603 |
| TigerBot | 182,249 | GPT-4all | 47,627 |
| COIG | 15,181 | LIMA | 309 |
| AlpacaDataCleaned | 13,805 | GPT-4-LLM | 9,978 |
| Bactrian-X | 3,313 | OpenOrca | 834,974 |

Table 2: The composition of the selected valuable data entries $\mathcal{V}$.

### 3.3 VALUABLE DATA ENTRIES CONVERSION

Following valuable data entries selection, we further convert the selected entries, allowing LLMs to learn how to invoke the external tools effectively within the proper context to support their reasoning process. In particular, we draw on previous methodologies Schick et al. (2024); Dubey et al. (2024); Thoppilan et al. (2022) by embedding special characters in each selected entry to enable the external tool invocation, and LLMs are used to pinpoint the appropriate context for calling external tools.

---

**Algorithm 1** Process Special Tokens During Inference

**Require:** Python interpreter and $inputs$.
**Ensure:** Execute the code enclosed by `<python>` and `</python>`, and use the captured output as a condition for the subsequent text generation.
1: Initialize an empty list $outputs$
2: Initialize $condition \leftarrow$ None
3: **while** $outputs$ is empty **or** $outputs[-1] \neq$ `<|end_of_text|>` **do**
4:     $output \leftarrow \text{LLM}(inputs)$
5:     **if** $output =$ `<python>` **then**
6:         $start \leftarrow$ length of $outputs$ + length of `<python>`
7:     **else if** $output =$ `</python>` **then**
8:         Extract substring $code \leftarrow outputs[start :]$
9:         $condition \leftarrow \text{ExecutePython}(code)$
10:     **end if**
11:     Append $output$ to both $inputs$ and $outputs$
12:     **if** $condition$ is not None **then**
13:         Append $condition$ to both $inputs$ and $outputs$
14:         Reset $condition \leftarrow$ None
15:     **end if**
16: **end while**
17: Post-process and return $outputs$

---

As illustrated in the examples in Figure 1, `<python>` and `</python>` are represented as a pair of special tokens. The content enclosed by the special tokens specifies the Python invocation for using external tools. During the construction of ToolBridge, we predominantly use GPT-4o-mini to insert the special tokens in the appropriate context within each data entry identified in Section 3.2, as well as to create the associated code for invoking external tools. To facilitate the return of tool execution

results, we examine the code generated by GPT-4o when calling Python API as part of its reasoning process. It is observed that the final results are always printed at the end of the code. In line with the strategy of GPT-4o, we include a directive in the prompt for GPT-4o-mini to print the final result of the tool execution as the concluding line of the code (refer to Appendix A.2).

Following GPT-4o-mini's processing of $\mathcal{V}$, we retrieve the code segments between `<python>` and `</python>`, execute them, and finally insert the captured output after corresponding `</python>`, where the output will be wrapped within another pair of special tokens, denoted as `<result>` and `</result>`. To summarize, the tool invocation in ToolBridge can be formatted as,

```
<python>generated code</python><result>captured output</result>.
```

During the reasoning process of the models, it is sufficient to verify the existence of the pre-defined special tokens `<python>` and `</python>`. If identified, the Python interpreter will be adopted to execute the enclosed code between the special tokens, and the final result is wrapped in `<result>` and `</result>` to condition further text generation. Algorithm 1 describes the primary inference process of LLMs post-SFT on the ToolBridge dataset.

Table 3 summarizes the composition of the data entries after converting $\mathcal{V}$, denoted as $\mathcal{C}$.

| Source | # of Entries | Source | # of Entries |
|---|---|---|---|
| School Math 0.25M | 150,104 | ChatAlpaca | 116 |
| Platypus | 82 | ShareGPT90K | 3 |
| WizardLM_Orca | 88 | WizardLM_evol_instruct_70k | 3,716 |
| WizardLM evolve_instruct V2 | 8 | OpenOrca | 101,715 |
| TigerBot | 66,793 | GPT-4all | 8,936 |
| COIG | 7,877 | LIMA | 107 |
| AlpacaDataCleaned | 4,510 | GPT-4-LLM | 2,577 |
| Bactrian-X | 84 | | |

Table 3: The composition of the converted data entries $\mathcal{C}$.

It is worth noting that the data entries in $\mathcal{C}$, totaling 364,605, are significantly fewer than those in $\mathcal{V}$ which amount to 1,527,153. We attribute this to the following factors,

- The returned data entries that lack the Python code inserted by GPT-4o-mini are eliminated. Approximately 19.2% of the data entries in $\mathcal{V}$ fall under this category, which indicates that GPT-4o-mini regards these entries as not requiring external tools to aid in LLMs' reasoning.

- The returned data format did not match the expected structure, resulting in parsing failure. Instances include an unequal count of `<python>` and `</python>` tokens, modifications to the original content alongside Python code insertion, to name a few. Such entries account for approximately 27.2% of the data entries in $\mathcal{V}$.

- The request to GPT-4o-mini fails. These data entries constitute approximately 2.1% of $\mathcal{V}$.

- With Python's *ast* library, we filter out the returned data entries where the inserted code is only an assignment followed by a print statement. Such entries represent about 4.8% of $\mathcal{V}$. We provide the detailed algorithm we use to filter such data entries in Appendix A.3.

- We filter around 22.8% data entries of $\mathcal{V}$ to remove tool calls that could not be executed or the execution time of the tool exceeded 30 seconds (refer to Appendix A.4 for details).

In summary, 23.9% of the entries from $\mathcal{V}$ remain in the converted data entries $\mathcal{C}$.

### 3.4 DATA ENTRIES FILTERING BY CONSISTENCY VALIDATION

In practice, we observe that LLMs trained on $\mathcal{C}$ do not always base their subsequent contents on the results produced by the yielded Python code during inference. So, we conduct a reassessment of the data entries within $\mathcal{C}$ and observe that the execution results from the code generated by GPT-4o-mini also does not always align with the ensuing text, which can explain LLMs' sporadic inconsistencies between tool execution results and further contents during inference.

To alleviate the issues above, we propose to filter out the entries in $\mathcal{C}$ where the tool execution results are inconsistent with the following text, which is accomplished by validating if the execution results are included in the subsequent content in our approach. Upon the conclusion of the filtering process, the open-source dataset ToolBridge is constructed. In Appendix A.5, we compare the generated text of Llama3-8B after SFT on $\mathcal{C}$ and ToolBridge, which demonstrates the necessity for the data entries filtering by consistency validation.

The data sources that comprise ToolBridge, totaling 178,023 entries, are outlined in Table 4, which represents 48.8% of the total data entries in $\mathcal{C}$.

| Source | # of Entries | Source | # of Entries |
|---|---|---|---|
| School Math 0.25M | 100,836 | ChatAlpaca | 17 |
| Platypus | 35 | ShareGPT90K | 3 |
| WizardLM_Orca | 29 | WizardLM_evol_instruct_70k | 794 |
| WizardLM evolve_instruct V2 | 1 | OpenOrca | 46,449 |
| TigerBot | 22,306 | GPT-4all | 2,616 |
| COIG | 2,706 | LIMA | 27 |
| AlpacaDataCleaned | 1,129 | GPT-4-LLM | 1,043 |
| Bactrian-X | 32 | | |

Table 4: The final composition of our proposed ToolBridge.

## 4 EXPERIMENTS

In this section, we first present the statistics of ToolBridge and then investigate whether LLMs with SFT on ToolBridge could leverage external tools to facilitate their reasoning process. The evaluation consists of two components: (1) by comparing LLMs' performance on standard benchmarks before and after SFT on ToolBridge (Section 4.3); (2) by evaluating the accuracy of the models on custom datasets RandomQA and FACT (Section 4.4).

| Source | # of Tool Use | # of Libraries | Source | # of Tool Use | # of Libraries |
|---|---|---|---|---|---|
| School Math 0.25M | 104,983 | 8 | ChatAlpaca | 73 | 1 |
| Platypus | 36 | 3 | ShareGPT90K | 3 | 0 |
| WizardLM_Orca | 33 | 4 | WizardLM_evol_instruct_70k | 836 | 22 |
| WizardLM evolve_instruct V2 | 1 | 0 | OpenOrca | 46,832 | 28 |
| TigerBot | 22,507 | 40 | GPT-4all | 2,870 | 42 |
| COIG | 2,719 | 28 | LIMA | 27 | 2 |
| AlpacaDataCleaned | 1,139 | 23 | GPT-4-LLM | 1,052 | 14 |
| Bactrian-X | 36 | 3 | | | |

Table 5: Statistics of the usage of external tools in ToolBridge.

### 4.1 DATASET STATISTICS OF TOOLBRIDGE

In Table 4, we present 15 source datasets involved in the data entries of ToolBridge, along with their respective composition ratios. To prevent any confusion for the datasets (*e.g*, other datasets with the same name and the same datasets with different versions), we also provide download links for these datasets in Appendix A.6.

Besides, we provide a summary of the frequency of external tool calls in ToolBridge and the variety of Python packages used for these calls, as presented in Table 5. By comparing Table 4 and Table 5, we can observe that the majority of the data entries in ToolBridge involve only a single external tool call. Furthermore, Table 5 also suggests that the data entries in ToolBridge originating from datasets with narrower topics (*e.g.*, School Math 0.25M) generally leverage fewer kinds of Python packages compared to those from datasets with broader topics (*e.g.*, TigerBot), which is consistent with logic. To summarize, there are 183,147 external tool calls in our ToolBridge dataset, utilizing a total of 60 Python packages, including *requests*, *math*, *datetime*, *sklearn*, to name a few.

In Appendix A.12, we also quantify the usage frequency of the 60 Python packages incorporated in ToolBridge, and their distribution is illustrated in Table 13.

## 4.2 EXPERIMENTAL SETUP

**Baseline Models.** Our experiments incorporate four baseline models: the base model of Mistral-7B, Llama2-7B, Llama3-8B and Llama3-70B. Also, we remove all the external tool invocation sections in each entry in ToolBridge (denote as ToolBridge$^§$) and report the accuracy of four baseline models SFT on ToolBridge$^§$ as four additional baseline models.

**Benchmark Datasets.** The standard benchmark datasets leveraged in our experiments include GSM 8K Cobbe et al. (2021), GSM Plus Li et al. (2024), MathBench Liu et al. (2024), Stanford WebQA Berant et al. (2013) and TruthfulQA Lin et al. (2021). For GSM 8k and GSM Plus, the performance is evaluated on their respective test sets under few-shot setting, where we leverage a fixed CoT-n-shot prompt template, as outlined in Li et al. (2024). For MathBench, we report results on MathBench-A, where we transform the multiple-choice questions in the College, High and Middle categories into a question-and-answer format for CoT-n-shot evaluation. To differentiate from standard MathBench, we refer to this adjusted dataset as MathBench$^*$.

We also develop two custom datasets, termed RandomQA and FACT, to evaluate the capabilities of LLMs in data processing, numerical computation and factual retrieval. Section 4.4 elaborates on the specifics of both datasets.

**SFT Settings.** All the models in our experiments are trained with the open-source TRL library from Hugging Face von Werra et al. (2020). The LoRA module Hu et al. (2021) is employed to carry out SFT on the base model of Mistral-7B, Llama2-7B, Llama3-8B and Llama3-70B using ToolBridge$^§$ or ToolBridge, with a LoRA rank of 16. Model training is conducted on $64 \times$ MI200 64GB GPUs, with each processing a batch size of 2 (*i.e.*, total batch size is 128). AdamW is employed to optimize the parameters of LoRA, with a cosine learning rate scheduler, configuring the initial lr at 3e-5 and the total training epoch at 3.

**Inference Settings.** The primary process of model inference is described in Algorithm 1. Moreover, to handle potential tool call failures during inference, we propose to eliminate failed tool calls from the current output before conditioning the generation of further text. All trained models are evaluated on $16 \times$ MI200 64 GB GPUs, with the max new tokens set to 512 and the temperature set to zero.

| Models | SFT data | GSM 8k | GSM Plus | MathBench$^*$ |
|---|---|---|---|---|
| Llama2-7B | - | 13.6 | 8.9 | 18.0 |
| Llama2-7B-Lora | ToolBridge$^§$ | 16.9 | 9.9 | 19.1 |
| Llama2-7B-Lora | ToolBridge | 18.1 | 11.0 | 21.4 |
| Llama3-8B | - | 52.3 | 36.9 | 33.0 |
| Llama3-8B-Lora | ToolBridge$^§$ | 53.4 | 37.8 | 35.2 |
| Llama3-8B-Lora | ToolBridge | 55.8 | 40.0 | 37.4 |
| Mistral-7B | - | 38.1 | 25.1 | 27.8 |
| Mistral-7B-Lora | ToolBridge$^§$ | 42.8 | 27.6 | 28.9 |
| Mistral-7B-Lora | ToolBridge | 45.0 | 29.8 | 31.0 |
| Llama3-70B | - | 75.3 | 54.4 | 42.1 |
| Llama3-70B-Lora | ToolBridge$^§$ | 78.5 | 57.6 | 44.1 |
| Llama3-70B-Lora | ToolBridge | 80.1 | 59.8 | 46.9 |

Table 6: Ablation studies on GSM 8k, GSM Plus and MathBench$^*$ with (8 shots, CoT) setting.

## 4.3 RESULTS ON STANDARD BENCHMARKS

In this section, we conduct ablation studies on standard benchmark datasets, including TruthfulQA, GSM 8k, GSM Plus, MathBench and Stanford WebQA, where GSM 8k, GSM Plus and MathBench are primarily responsible for evaluating the capability of LLMs in numerical reasoning and computation, and TruthfulQA and Stanford WebQA are primarily adopted to assess the ability of LLMs in factual retrieval. Table 6, 7 and 8 demonstrates the evaluation results.

It is observed that the models SFT on ToolBridge significantly outperform the baseline models. For example, Llama3-8B SFT on ToolBridge brings 3.1% and 2.2% accuracy improvements to the base model of Llama3-8B and the Llama3-8B model SFT on ToolBridge$^§$ when evaluating on GSM Plus, respectively. These results indicate that SFT on ToolBridge can help strengthen LLMs' capabilities in handling numerical computations. Furthermore, ToolBridge is able to facilitate abilities of LLMs

| Models | SFT data | Stanford WebQA |
|---|---|---|
| Llama3-8B | - | 21.2 |
| Llama3-8B-Lora | ToolBridge[§] | 37.7 |
| Llama3-8B-Lora | ToolBridge | 39.9 |
| Mistral-7B | - | 34.4 |
| Mistral-7B-Lora | ToolBridge[§] | 35.8 |
| Mistral-7B-Lora | ToolBridge | 39.1 |

Table 7: Ablation studies on Stanford WebQA under zero-shot setting.

| Models | SFT data | ROUGE1 | BLEURT |
|---|---|---|---|
| Llama3-8B | - | 41.2 | 34.6 |
| Llama3-8B-Lora | ToolBridge[§] | 47.0 | 42.8 |
| Llama3-8B-Lora | ToolBridge | 48.7 | 44.4 |
| Mistral-7B | - | 43.5 | 39.4 |
| Mistral-7B-Lora | ToolBridge[§] | 44.9 | 42.3 |
| Mistral-7B-Lora | ToolBridge | 47.7 | 44.9 |

Table 8: Ablation studies on TruthfulQA under zero-shot setting.

in factual retrieval. As shown in Table 7 and 8, SFT on ToolBridge enables Llama3-8B and Mistral-7B to achieve notable gains on Stanford WebQA and TruthfulQA. Specifically, ToolBridge increases the accuracy of Llama3-8B on Stanford WebQA from 21.2% to 39.9%, and on TruthfulQA, it boosts ROUGE1 from 41.2% to 48.7% and BLEURT from 34.6% to 44.4%.

The results above demonstrate that LLMs can effectively learn how to use external tools to enhance their capabilities in basic functions after SFT on ToolBridge. Moreover, it is worthy noting that there is considerable room for improvements in these results, as our emphasis is on the training data, with minimal adjustments made to the model architectures and training strategies, which may help LLMs better learn how to employ external tools through ToolBridge.

| Models | SFT data | RandomQA-DP-B1 | RandomQA-DP-B2 | RandomQA-NC-B1 | RandomQA-NC-B2 |
|---|---|---|---|---|---|
| Llama2-7B | - | 10.0 | 9.0 | 3.3 | 3.2 |
| Llama2-7B-Lora. | ToolBridge[§] | 19.2 | 16.6 | 7.7 | 8.6 |
| Llama2-7B-Lora. | ToolBridge | 53.2 | 54.0 | 63.4 | 60.7 |
| Llama3-8B | - | 9.6 | 9.2 | 5.8 | 7.0 |
| Llama3-8B-Lora | ToolBridge[§] | 30.3 | 29.0 | 15.8 | 13.9 |
| Llama3-8B-Lora | ToolBridge | 62.1 | 60.0 | 82.1 | 80.1 |
| Mistral-7B | - | 10.8 | 9.0 | 13.8 | 13.6 |
| Mistral-7B-Lora | ToolBridge[§] | 24.7 | 23.2 | 16.8 | 16.5 |
| Mistral-7B-Lora | ToolBridge | 61.8 | 60.5 | 83.3 | 82.5 |
| Llama3-70B | - | 20.0 | 17.1 | 9.6 | 8.9 |
| Llama3-70B-Lora | ToolBridge[§] | 32.1 | 31.7 | 22.0 | 20.3 |
| Llama3-70B-Lora | ToolBridge | 74.2 | 69.9 | 89.7 | 89.1 |

Table 9: Experimental results on RandomQA under zero-shot setting, where DP denotes data processing and NC means numerical computation.

## 4.4 RESULTS ON CUSTOM BENCHMARKS

To further assess whether SFT on the ToolBridge dataset can equip LLMs with the ability to leverage external tools for aiding its reasoning process, we propose to design two custom datasets to evaluate LLMs' performance before and after SFT on the ToolBridge dataset.

**RandomQA.** To assess LLMs' accuracy in data processing and numerical computation capabilities after SFT on ToolBridge, we propose to design 30 templates capable of generating question-answer pairs to validate the abilities of LLMs in data processing and numerical computations, respectively. Here is one example,

```
# Template1: Reverse the order of elements in a list
array = [random.randint(1, 10000) for _ in range(random.randint(5, 15))]
question = f"Reverse the order of the elements in the list {array} and
    ↪  then plus 3 for each element."
answer = array[::-1]
answer = [a + 3 for a in answer]
```

The complete list can be found in Appendix A.7. With these pre-defined templates, we first generate four RandomQA datasets, each consisting of 1,000 data entries, and focusing on data processing or numerical computation, *i.e.*, RandomQA-DP-B1/2, RandomQA-NC-B1/2, where DP signifies using template related to data processing to yield the dataset and NC means numerical computation. Then, we evaluate the accuracy of LLMs on the four datasets before and after SFT on ToolBridge. Table 9 demonstrates the results. It is observed that after SFT on ToolBridge, the models shows a significant increase in accuracy on RandomQA.

| Models | SFT data | FACT-200-Batch1 | FACT-200-Batch2 | FACT-200-Batch3 |
|---|---|---|---|---|
| Llama2-7B | - | 69.5 | 55.0 | 49.0 |
| Llama2-7B-Lora | ToolBridge$^{\S}$ | 86.0 | 67.5 | 65.7 |
| Llama2-7B-Lora | ToolBridge | 88.5 | 72.5 | 73.2 |
| Llama3-8B | - | 79.0 | 60.5 | 62.2 |
| Llama3-8B-Lora | ToolBridge$^{\S}$ | 89.0 | 73.0 | 73.7 |
| Llama3-8B-Lora | ToolBridge | 90.0 | 73.5 | 80.2 |
| Mistral-7B | - | 85.0 | 67.5 | 65.9 |
| Mistral-7B-Lora | ToolBridge$^{\S}$ | 86.5 | 70.0 | 66.2 |
| Mistral-7B-Lora | ToolBridge | 90.5 | 72.0 | 77.3 |
| Llama3-70B | - | 76.0 | 53.5 | 54.0 |
| Llama3-70B-Lora | ToolBridge$^{\S}$ | 88.3 | 72.4 | 70.7 |
| Llama3-70B-Lora | ToolBridge | 91.2 | 74.6 | 82.6 |

Table 10: Experimental results on FACT under zero-shot setting.

**FACT.** To determine if the factual retrieval skills of LLMs can be improved by SFT on ToolBridge, we construct the FACT datasets. Specifically, we begin by prompting GPT-4o to produce thousands of question-answer pairs focused on factual retrieval. One example prompt is as following,

```
1. Generate 100 Q&A pairs for LLM factual retrieval testing. The
↪  question topic should be related with Geography. Return them
↪  as a Python dictionary, with concise answers (3-5 words).
```

Appendix A.8 contains the entire set of the adopted prompts for constructing FACT. Upon obtaining the candidate question-answer pairs, we continuously draw random entries from them and manually check their correctness until 200 correct data entries are verified. Through iterating the above process three times with five different prompts each time, we construct three FACT datasets, each with a size of 200 entries, termed FACT-200-Batch1/2/3. Table 10 presents a comparison of the performance of LLMs on the three datasets before and after SFT on the ToolBridge dataset, which shows that LLMs demonstrates enhanced abilities in retrieving factual content.

Besides, one should be aware that relying on external tools to improve the fact retrieval performance of LLMs can be inconsistent. During our experiments, we uncover the following deficiencies

- Using external tools like the *requests* module to obtain information from external websites can lead to potentially unreliable information and LLMs thus yield incorrect predictions. A solution to address this issue is to construct a curated list of trustworthy domains, restricting LLMs from accessing websites beyond this approved whitelist. Besides, we should further refine the training entries to exclude unreliable access links.
- The content scraped from web pages may be overly lengthy, hindering the ability of LLMs to efficiently extract useful information from the scraped contents.
- When LLMs access some external websites, they may attempt to download files from those sites, which poses potential security risks in real-world applications.

Therefore, we recommend that ToolBridge is only used for academic research, with controlled measures in place when it attempts to access external websites.

## 5 CONCLUSION

This paper proposes ToolBridge, which is the first open-source dataset for training LLMs in the use of external tools in the area of tool former. The principal steps to build ToolBridge involves valuable data entries selection, conversion and filtering. Ablation studies on standard and custom benchmark datasets demonstrate the effectiveness of ToolBridge.

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

# A APPENDIX

## A.1 LLAMA3-70B PROMPTS

The prompt applied while using Llama3-70B to identify valuable data is as follow,

```
Your task is to determine whether you can add calls to a Python API to a piece
↪  of text. The calls should help you get information required to complete the
↪  text. You only need to respond with "Yes" or "No", "Yes" means you can and
↪  "No" means you can't. Here are some examples:

Input:

{"messages": [{"role": "user", "content": "Sort the numbers in the list arr =
↪  [1, 10, 2, 5, -2, 11, 12] in descending order."}, {"role": "assistant",
↪  "content": "The sorted list in descending order is: [12, 11, 10, 5, 2, 1,
↪  -2]."}]}

Output:

Yes

Input:

{"messages": [{"role": "user", "content": "Can you tell me a little bit about
↪  what LaTeX is?"}, {"role": "assistant", "content": "LaTeX is a high-quality
↪  typesetting system; it includes features designed for the production of
↪  technical and scientific documentation. LaTeX is the de facto standard for
↪  the communication and publication of scientific documents. It is widely used
↪  by mathematicians, scientists, engineers, philosophers, linguists,
↪  economists, and other scholars in academia and the professional world."}]}

Output:

No

Input:

{"messages": [{"role": "user", "content": "What is the value of sin 40 degrees
↪  plus cos 31 degrees?"}, {"role": "assistant", "content": "The value is
↪  approximately sin 40 + cos 31 = 0.6428 + 0.8572 = 1.500."}]}

Output:

Yes

Input:

{"messages": [{"role": "user", "content": "Write a Python script that reads an
↪  image and recognizes the text on it."}, {"role": "assistant", "content": "To
↪  read an image and recognize text on it in Python, you can use the
↪  pytesseract library along with Pillow for image processing. Here's a simple
↪  example: \nimport pytesseract; from PIL import Image;
↪  print(pytesseract.image_to_string(Image.open('path_to_image.jpg'))). \nMake
↪  sure you have Tesseract OCR installed on your machine and the required
↪  libraries (pytesseract and Pillow) installed in your Python environment. You
↪  can install them using: \npip install pytesseract pillow"}]}

Output:

No

Input:

PLACEHOLDER

Output:
```

## A.2 GPT-4O-MINI PROMPTS

The prompt applied while using GPT-4o-mini to convert data entries is as follow,

```
Your task is to add calls to a Python API to a piece of text. The calls should help you
↪ get information required to complete the text. You can call the API by writing
↪ "<python>code</python>" where "code" is the code to be executed. The last line of
↪ all code should print the variable that stores the final result. Here are some
↪ examples of API calls:

Input:

{"messages": [{"role": "user", "content": "Which number is greater, 13.11 or 13.8?"},
↪ {"role": "assistant", "content": "13.8 is greater than 13.11."}]}

Output:

{"messages": [{"role": "user", "content": "Which number is greater, 13.11 or 13.8?"},
↪ {"role": "assistant", "content": "<python>greater_number = max(13.11,
↪ 13.8)\nprint(greater_number)</python> 13.8 is greater than 13.11."}]}

Input:

{"messages": [{"role": "user", "content": "How many unique words are there in the
↪ sentence 'The quick brown fox jumps over the lazy dog'?"}, {"role": "assistant",
↪ "content": "There are eight unique words in the sentence 'The quick brown fox jumps
↪ over the lazy dog.'"}]}

Output:

{"messages": [{"role": "user", "content": "How many unique words are there in the
↪ sentence 'The quick brown fox jumps over the lazy dog'?"}, {"role": "assistant",
↪ "content": "There are <python>unique_words = len(set('The quick brown fox jumps over
↪ the lazy dog'.lower().split()))\nprint(unique_words)</python> eight unique words in
↪ the sentence 'The quick brown fox jumps over the lazy dog.'"}]}

Input:

{"messages": [{"role": "user", "content": "What is the area of a circle with a radius of
↪ 5?"}, {"role": "assistant", "content": "The area of a circle with radius 5 is
↪ 78.54."}]}

Output:

{"messages": [{"role": "user", "content": "What is the area of a circle with a radius of
↪ 5?"}, {"role": "assistant", "content": "The area of a circle with radius 5 is
↪ <python>import math\narea = math.pi * 5**2\nprint(area)</python> 78.54."}]}

Input:

{"messages": [{"role": "user", "content": "Sort the numbers [5, 3, 8, 1, 2] in ascending
↪ order."}, {"role": "assistant", "content": "The sorted list is [1, 2, 3, 5, 8]."}]}

Output:

{"messages": [{"role": "user", "content": "Sort the numbers [5, 3, 8, 1, 2] in ascending
↪ order."}, {"role": "assistant", "content": "The sorted list is <python>lst =
↪ sorted([5, 3, 8, 1, 2])\nprint(lst)</python> [1, 2, 3, 5, 8]."}]}

Input:

{"messages": [{"role": "user", "content": "Extract the domain from the email
↪ 'example@test.com'."}, {"role": "assistant", "content": "The domain of the email
↪ 'example@test.com' is 'test.com'."}]}

Output:

{"messages": [{"role": "user", "content": "Extract the domain from the email
↪ 'example@test.com'."}, {"role": "assistant", "content": "The domain of the email
↪ 'example@test.com' is <python>domain =
↪ 'example@test.com'.split('@')[1]\nprint(domain)</python> 'test.com'."}]}

Input:

PLACEHOLDER

Output:
```

## A.3 DETECTING SIMPLE CODE STRUCTURES: ASSIGNMENT FOLLOWED BY PRINT STATEMENT

We use the following code to determine if the inserted code in each entry is simply an assignment followed by printing the variable.

**Algorithm 2** Identifying Code Patterns: Distinguishing Assignments Followed by Print Statements

```python
import ast

'''
Function:
    Identifying Code Patterns: Distinguishing Assignments Followed
        by Print Statements
Argument:
    node: node = ast.parse(code)
'''
def isuselesscode(node):
    # Check if the node is an AST Module
    if isinstance(node, ast.Module):
        # Ensure the module has exactly two statements:
        # an assignment and an expression.
        if len(node.body) == 2 and isinstance(node.body[0], ast.
            Assign) and isinstance(node.body[1], ast.Expr):
            assign_node = node.body[0]
            expr_node = node.body[1]
            # Check if the assignment targets a variable and
            # the value is a constant.
            if isinstance(assign_node.targets[0], ast.Name) and
                isinstance(assign_node.value, ast.Constant):
                # Check if the expression is a function call
                # to 'print'.
                if isinstance(expr_node.value, ast.Call) and
                    isinstance(expr_node.value.func, ast.Name) and
                    expr_node.value.func.id == 'print':
                    # Ensure 'print' has exactly one argument.
                    if len(expr_node.value.args) == 1:
                        arg = expr_node.value.args[0]
                        # Check if the argument to 'print' is
                        # the same variable assigned earlier.
                        if isinstance(arg, ast.Name) and arg.id ==
                            assign_node.targets[0].id:
                            return True
                        # Alternatively, check if 'print' uses an
                        # f-string format with the variable.
                        elif isinstance(arg, ast.JoinedStr):
                            for value in arg.values:
                                if isinstance(value, ast.
                                    FormattedValue) and isinstance(
                                    value.value, ast.Name):
                                    # Confirm the formatted
                                    # variable is the same as
                                    # the assigned variable.
                                    if value.value.id ==
                                        assign_node.targets[0].id:
                                        return True
    return False
```

## A.4 OPTIMIZING DATASET QUALITY BY REMOVING NON-EXECUTABLE AND SLOW TOOL CALLS

The core code for removing data entries that either fail to execute or exceed the execution time limit is as follows.

**Algorithm 3** Efficient Dataset Curation: Filtering Non-Executable and Time-Consuming Tool Calls

```python
import re
import io
import contextlib
import multiprocessing

# Execute Python code
def executecode(code):
    with io.StringIO() as buf, contextlib.redirect_stdout(buf):
        try:
            exec(code)
            return buf.getvalue().strip()
        except Exception:
            return None

# Execute Python code with timeout = 30 second
def safeexecutecode(code, timeout=30):
    result_queue = multiprocessing.Queue()
    def target():
        result = executecode(code)
        result_queue.put(result)
    process = multiprocessing.Process(target=target)
    process.start()
    process.join(timeout)
    if process.is_alive():
        process.terminate()
        process.join()
        return None
    else:
        return result_queue.get() if not result_queue.empty() else
            None

# Filtering
DATASET_FILTERED = []
for item in tqdm(DATASET):
    success_calling = False
    for message in item['messages']:
        if message['role'] == 'assistant':
            code_matches = re.findall(r'<python>(.*?)</python>',
                message['content'], re.DOTALL)
            if code_matches:
                for code in code_matches:
                    python_str = f'<python>{code}</python>'
                    result = safeexecutecode(code=code)
                    if result is None:
                        message['content'] = message['content'].
                            replace(python_str, '')
                    else:
                        result_str = f'<result>{result}</result>'
                        message['content'] = message['content'].
                            replace(python_str, f'{python_str}{
                            result_str}')
                        success_calling = True
    if success_calling:
        DATASET_FILTERED.append(item)
```

## A.5   COMPARE OUTPUTS FROM LLAMA3 TRAINED ON $\mathcal{C}$ AND TOOLBRIDGE

We notice in actual applications that the models train on $\mathcal{C}$ may not always use the Python output as a basis for generating further content. Here are some examples.

---

***Inconsistent outputs from Llama3 trained on $\mathcal{C}$***

*Example 1:*

**User:** "Count the number of vowels in the sentence 'This is a simple sentence'."

**Assistant**: "There are <python>sentence = 'This is a simple sentence'\nvowels = 'aeiou'\ncount = sum(1 for char in sentence if char.lower() in vowels)\nprint(count)</python><result>8</result> 5 vowels in the sentence 'This is a simple sentence'."

*Example 2:*

**User:** "What is the date 30 days after January 31, 2022?"

**Assistant**: "The date 30 days after January 31, 2022 is <python>from datetime import datetime, timedelta\n\ndate = datetime.strptime('2022-01-31', '%Y-%m-%d')\nnew_date = date + timedelta(days=30)\nresult = new_date.strftime('%B %d, %Y')\nprint(result)</python><result>March 02, 2022</result> March 1, 2022. Note that February only has 28 days in non-leap years, so the 30th day would fall on March 1, 2022 in a leap year. "

---

Figure 3: Inconsistency between LLMs' outputs and the returned results from Python interpreter.

By filtering out the entries in $\mathcal{C}$ that show inconsistencies between the Python execution results and the following content, we create the ToolBridge dataset. This refined dataset is then used to train the Llama3 model, yielding outputs consistent with the Python results. Here are some examples.

---

***Consistent outputs from Llama3 trained on ToolBridge***

*Example 1:*

**User:** "Count the number of vowels in the sentence 'This is a simple sentence'."

**Assistant**: "There are <python>sentence = 'This is a simple sentence'\nvowels = ['a', 'e', 'i', 'o', 'u']\ncount = sum(1 for char in sentence.lower() if char in vowels)\nprint(count)</python><result>8</result> 8 vowels in the sentence 'This is a simple sentence'."

*Example 2:*

**User:** "What is the date 30 days after January 31, 2022?"

**Assistant**: "The date 30 days after January 31, 2022 is <python>from datetime import datetime, timedelta\n\ndate = datetime.strptime('2022-01-31', '%Y-%m-%d')\nnew_date = date + timedelta(days=30)\nresult = new_date.strftime('%B %d, %Y')\nprint(result)</python><result>March 02, 2022</result> February 30, 2022. However, this date is not valid as February only has 28 days in a non-leap year. So, the date would be March 02, 2022."

---

Figure 4: Consistency between LLMs' outputs and the returned results from Python interpreter.

A.6 Links to data sources used in the dataset pool

To avoid any confusion regarding the datasets, the download links for each dataset are also provided in Table 11. Note that, links to all datasets from the dataset pool $\mathcal{P}$ are presented in the table, even those not incorporated into ToolBridge.

| Dataset | Source Link |
|---|---|
| School Math 0.25M | https://huggingface.co/datasets/BelleGroup/school_math_0.25M |
| ChatAlpaca | https://github.com/cascip/ChatAlpaca |
| Platypus | https://huggingface.co/datasets/garage-bAInd/Open-Platypus |
| ShareGPT90K | https://huggingface.co/datasets/liyucheng/ShareGPT90K |
| WizardLM_Orca | https://huggingface.co/datasets/pankajmathur/WizardLM_Orca |
| WizardLM_evol_instruct_70k | https://huggingface.co/datasets/WizardLMTeam/WizardLM_evol_instruct_70k |
| WizardLM evolve_instruct V2 | https://huggingface.co/datasets/WizardLMTeam/WizardLM_evol_instruct_V2_196k |
| OpenOrca | https://huggingface.co/datasets/Open-Orca/OpenOrca |
| TigerBot | https://huggingface.co/datasets/TigerResearch/sft_en, TigerResearch/sft_zh |
| GPT-4all | https://huggingface.co/datasets/nomic-ai/gpt4all-j-prompt-generations |
| COIG | https://huggingface.co/datasets/BAAI/COIG |
| LIMA | https://huggingface.co/datasets/GAIR/lima |
| AlpacaDataCleaned | https://huggingface.co/datasets/yahma/alpaca-cleaned |
| GPT-4-LLM | https://github.com/Instruction-Tuning-with-GPT-4/GPT-4-LLM |
| Bactrian-X | https://huggingface.co/datasets/MBZUAI/Bactrian-X |
| code_instructions_120k_alpaca | https://huggingface.co/datasets/iamtarun/code_instructions_120k_alpaca |
| TSI-v0 | https://huggingface.co/datasets/tasksource/tasksource-instruct-v0 |
| Alpaca | https://github.com/tatsu-lab/stanford_alpaca |
| No Robots | https://huggingface.co/datasets/HuggingFaceH4/no_robots |
| Baize | https://github.com/project-baize/baize-chatbot |
| LaMini-Instruction | https://huggingface.co/datasets/MBZUAI/LaMini-instruction |
| tiny-codes | https://huggingface.co/datasets/nampdn-ai/tiny-codes |
| self-instruct | https://github.com/yizhongw/self-instruct |
| ign_clean_instruct_dataset_500k | https://huggingface.co/datasets/ignmilton/ign_clean_instruct_dataset_500k |
| MOSS SFT | https://github.com/OpenMOSS/MOSS |

Table 11: Source links for the datasets utilized in the dataset pool.

Here is a brief description of each dataset,

- School Math 0.25M: It includes approximately 250,000 Chinese math problems generated by the BELLE project, along with their solution processes.

- ChatAlpaca: ChatAlpaca is a comprehensive dataset created to assist researchers in building models for instruction-following across multi-turn conversations. It expands upon the Stanford Alpaca dataset by incorporating a broader range of multi-turn instructions and their corresponding responses.

- Platypus: This dataset is designed to enhance the logical reasoning capabilities of LLMs and was instrumental in training the Platypus2 models. It combines eleven public datasets, carefully curated through keyword filtering and further refined using Sentence Transformers to exclude questions with more than 80% similarity.

- ShareGPT90K: A high quality dataset generated by using GPT-4.

- WizardLM_Orca: Enhanced WizardLM dataset, generated using the Orca methodology.

- WizardLM_evol_instruct_70k: This is the training data of WizardLM.

- WizardLM evolve_instruct V2: The dataset contains 143K mixed evolved data derived from Alpaca and ShareGPT. It represents the latest optimized version of Evol-Instruct training data for the WizardLM model.

- OpenOrca: A collection of augmented FLAN data, generated using the methodology described in the Orca paper.

- TigerBot: Datasets used to train TigerBot include pretraining data, STF data, and domain-specific datasets such as financial research reports.

- GPT-4all: A curated mix of subsets from OIG, P3, and StackOverflow, focusing on topics such as general question-answering and customized creative queries.

- COIG: A Chinese-focused dataset encompassing domains such as general-purpose QA, Chinese exams, and coding. Its quality has been verified by human annotators.

- LIMA: High quality SFT dataset used by LIMA.

- AlpacaDataCleaned: An improved and cleaned iteration of the Alpaca, GPT_LLM, and GPTeacher datasets.

- GPT-4-LLM: It is generated using GPT-4 and other LLMs to produce improved pairs and data for RLHF.

- Bactrian-X: A multilingual adaptation of the Alpaca and Dolly-15K datasets.

- code_instructions_120k_alpaca: Code instruction data formatted for instruction fine-tuning.

- TSI-v0: A multi-task instruction-tuning dataset derived from 475 Tasksource datasets, designed in a manner similar to the Flan and Natural Instructions datasets.

- Alpaca: It consists of 52K instruction-following examples, specifically designed for fine-tuning the Alpaca model.

- No Robots: High-quality, human-generated STF data in a single-turn format.

- Baize: A dialogue dataset generated by GPT-4 through self-talking, with questions and topics sourced from Quora, StackOverflow, and various medical knowledge bases.

- LaMini-Instruction: A dataset distilled from the FLAN collection, P3, and Self-Instruct.

- tiny-codes: This synthetic dataset comprises 1.6 million concise and clear code snippets, designed to help LLM models develop reasoning skills in both natural and programming languages. The dataset spans a wide range of programming languages, including Python, TypeScript, JavaScript, Ruby, Julia, Rust, C++, Bash, Java, C#, and Go.

- self-instruct: This dataset is generated using the methodology outlined in Self-Instruct: Aligning Language Models with Self-Generated Instructions.

- ign_clean_instruct_dataset_500k: A large-scale SFT dataset synthetically generated from a subset of Ultrachat prompts.

- MOSS SFT: A conversational dataset curated and developed by the MOSS team, with each entry annotated with labels for usefulness, loyalty, and harmlessness.

## A.7 Templates for Yielding RandomQA Dataset

The complete list of templates used to generate the RandomQA dataset is as follows,

```python
'''RandomQAGenerator'''
class RandomQAGenerator():
    question_types_data_processing = [
        "Sort an array in ascending order",
        "Transpose a 2D matrix",
        "Reverse the string",
        "Extract first N elements in a list",
        "Reverse the order of elements in a list",
        "Count the frequency of one character in a string",
        "Find the intersection of two strings",
        "Find the length of the longest word in a string",
        "Count the number of vowels in a string",
        "Convert a list of Celsius temperatures to Fahrenheit",
        "Calculate time difference between two time zones",
        "Find the leap year after a year",
        "Find the most common word in a paragraph",
        "Find the first recurring word in a string",
        "Extract all the numbers in a string",
        "Convert a decimal number to its binary equivalent",
        "Calculate the difference between two lists",
        "Find out all the numbers that are not unique",
        "Flatten a 2D list into a 1D list",
        "Remove duplicates from a list",
        "Filter elements in a list based on a condition",
        "Merge two dictionaries into one",
        "Extract all words of a specific length from a text",
        "Extract email addresses from a text",
        "Sort a list of strings by their length",
        "Check if two strings are anagrams",
        "Extract hashtags from a social media post",
        "Capitalize each word in a string",
        "Find the index of a substring in a string",
        "Replace all vowels in a string with a specific character",
    ]
    question_types_numerical_computation = [
        "Calculate the average of an array",
        "Find the maximum and minimum values of an array",
        "Calculate the dot product of two arrays",
        "Generate a set of random integers and find their sum",
        "Generate the smallest prime number greater than x",
        "Calculate the standard deviation of a list of floating-point
        ↪  numbers",
        "Generate a random matrix and find its inverse",
        "Find the median of an array",
        "Generate Fibonacci sequence up to n-th term",
        "Find the GCD (Greatest Common Divisor) of two numbers",
        "Calculate the factorial of a number",
        "Find the mode of a list of numbers",
        "Calculate the sum of even numbers in a list",
        "Calculate the cumulative sum of an array",
        "Calculate cosine value",
        "Square every number in a list",
        "Calculate the sum of squares of numbers in an array",
        "Find the n-th smallest number in an array",
        "Calculate the Euclidean distance between two points in a plane",
        "Calculate the compound interest given principal, rate, and
        ↪  time",
        "Calculate the perimeter of a rectangle given its length and
        ↪  width",
        "Sum all the digits of a given number",
```

```python
        "Calculate the area of a triangle given its base and height",
        "Find the real roots of a quadratic equation",
        "Calculate the sum of the cubes of a list",
        "Round all elements in a list to two decimal places",
        "Calculate the hypotenuse of a right triangle given the other two
        ↪   sides",
        "Sum all odd numbers in a list",
        "Generate the smallest N primes",
        "Find the sum of all elements above the main diagonal of a
        ↪   matrix"
    ]
    def __init__(self, num_gen_qa=1000):
        self.num_gen_qa = num_gen_qa
    '''generate'''
    def generate(self):
        qa_pairs = []
        for _ in range(self.num_gen_qa):

            ↪   qa_pairs.append(self.randomgenone(self.question_types_data_processing))
        pickle.dump(qa_pairs,
        ↪   open(f'random_qa_dp_{int(time.time())}.pkl', 'wb'))
        time.sleep(1)
        qa_pairs = []
        for _ in range(self.num_gen_qa):

            ↪   qa_pairs.append(self.randomgenone(self.question_types_data_processing))
        pickle.dump(qa_pairs,
        ↪   open(f'random_qa_dp_{int(time.time())}.pkl', 'wb'))
        time.sleep(1)
        qa_pairs = []
        for _ in range(self.num_gen_qa):

            ↪   qa_pairs.append(self.randomgenone(self.question_types_numerical_computation))
        pickle.dump(qa_pairs,
        ↪   open(f'random_qa_nc_{int(time.time())}.pkl', 'wb'))
        time.sleep(1)
        qa_pairs = []
        for _ in range(self.num_gen_qa):

            ↪   qa_pairs.append(self.randomgenone(self.question_types_numerical_computation))
        pickle.dump(qa_pairs,
        ↪   open(f'random_qa_nc_{int(time.time())}.pkl', 'wb'))
    '''randomgenone'''
    def randomgenone(self, question_types):
        # randomly choose a question type
        question_type = random.choice(question_types)
        # generate question and answer based on type
        # 1. Calculate the average of an array
        if question_type == "Calculate the average of an array":
            array = [round(random.uniform(-10000, 10000)) for _ in
            ↪   range(random.randint(5, 15))]
            question = f"Calculate the average of the array {array} and
            ↪   round the result to two decimal places."
            answer = round(sum(array) / len(array), 2)
        # 2. Find the maximum and minimum values of an array
        elif question_type == "Find the maximum and minimum values of an
        ↪   array":
            array = [round(random.uniform(-10000, 10000)) for _ in
            ↪   range(random.randint(5, 15))]
            max_or_min = random.choice(['maximum', 'minimum'])
            question = f"Find the {max_or_min} value of the array
            ↪   {array}, give the result of multiplying it by 7."
            answer = max(array) if max_or_min == 'maximum' else
            ↪   min(array)
            answer = answer * 7
```

```python
        # 3. Calculate the dot product of two arrays
        elif question_type == "Calculate the dot product of two arrays":
            length = random.randint(5, 15)
            array1 = [random.randint(20, 1000) for _ in range(length)]
            array2 = [random.randint(20, 1000) for _ in range(length)]
            question = f"Calculate the dot product of the arrays {array1}
                and {array2}."
            answer = sum(x * y for x, y in zip(array1, array2))
        # 4. Sort an array in ascending order
        elif question_type == "Sort an array in ascending order":
            array = [random.randint(-10000, 10000) for _ in
                range(random.randint(5, 15))]
            question = f"Sort the array {array} in ascending order."
            answer = sorted(array)
        # 5. Generate a set of random integers and find their sum
        elif question_type == "Generate a set of random integers and find
            their sum":
            array = [random.randint(1000, 100000) for _ in
                range(random.randint(5, 15))]
            question = f"Here is a set of random integers {array}, please
                find their sum."
            answer = sum(array)
        # 6. Generate the smallest prime number greater than x
        elif question_type == "Generate the smallest prime number greater
            than x":
            num = random.randint(2000, 100000)
            question = f"Generate the smallest prime number greater than
                {num}."
            answer = nextprime(num)
        # 7. Calculate the standard deviation of a list of floating-point
            numbers
        elif question_type == "Calculate the standard deviation of a list
            of floating-point numbers":
            array = [round(random.uniform(10, 1000), 2) for _ in
                range(random.randint(5, 15))]
            mean = sum(array) / len(array)
            variance = sum((x - mean) ** 2 for x in array) / len(array)
            question = f"Calculate the standard deviation of the array
                {array} and round the result to two decimal places."
            answer = round(variance ** 0.5, 2)
        # 8. Generate a random matrix and find its inverse
        elif question_type == "Generate a random matrix and find its
            inverse":
            matrix_len = random.randint(2, 10)
            matrix = [[random.randint(1, 1000) for _ in
                range(matrix_len)] for _ in range(matrix_len)]
            question = f"Here is a random matrix {matrix}, please find
                its inverse, you can answer with 'not invertible' if its
                inverse does not exist."
            det = np.linalg.det(matrix)
            if int(det) != 0:
                inv_matrix = np.linalg.inv(matrix).tolist()
            else:
                inv_matrix = "not invertible"
            answer = inv_matrix
        # 9. Count the frequency of one character in a string
        elif question_type == "Count the frequency of one character in a
            string":
            char = random.choice('abcdefghijklmnopqrstuvwxyz')
            string = ''.join(random.choices('abcdefghijklmnopqrstuvwxyz',
                k=random.randint(50, 100))) + char * 101
            question = f"Count the frequency of character {char} in the
                string '{string}'."
            answer = string.count(char)
        # 10. Square every number in a list
```

```python
        elif question_type == "Square every number in a list":
            array = [random.randint(1, 10000) for _ in
            ↪   range(random.randint(5, 15))]
            question = f"Square every number in the list {array}."
            answer = [x ** 2 for x in array]
        # 11. Find the median of an array
        elif question_type == "Find the median of an array":
            array = [random.randint(200000, 10000000) for _ in
            ↪   range(random.randint(5, 15))]
            sorted_array = sorted(array)
            question = f"Find the median of the array {array}, give the
            ↪   result of multiplying it by 9."
            answer = sorted_array[len(sorted_array) // 2]
            answer = answer * 9
        # 12. Generate Fibonacci sequence up to n-th term
        elif question_type == "Generate Fibonacci sequence up to n-th
        ↪   term":
            n = random.randint(5, 20)
            question = f"Generate the Fibonacci sequence up to the {n}-th
            ↪   term."
            fib = [0, 1]
            for i in range(2, n):
                fib.append(fib[-1] + fib[-2])
            answer = fib
        # 13. Transpose a 2D matrix
        elif question_type == "Transpose a 2D matrix":
            matrix_len = random.randint(2, 10)
            matrix = [[random.randint(-1000, 1000) for _ in
            ↪   range(matrix_len)] for _ in range(matrix_len)]
            question = f"Transpose the matrix {matrix}."
            answer = [list(row) for row in zip(*matrix)]
        # 14. Reverse the string
        elif question_type == "Reverse the string":
            string = ''.join(random.choices('abcdefghijklmnopqrstuvwxyz',
            ↪   k=random.randint(10, 20)))
            question = f"Reverse the string {string}, and splice it
            ↪   behind the string 'appleiphone'."
            answer = 'appleiphone' + string[::-1]
        # 15. Find the GCD (Greatest Common Divisor) of two numbers
        elif question_type == "Find the GCD (Greatest Common Divisor) of
        ↪   two numbers":
            answer = 0
            while answer <= 100:
                a, b = random.randint(200, 1000000), random.randint(200,
                ↪   1000000)
                question = f"Find the GCD of the numbers {a} and {b}."
                answer = math.gcd(a, b)
        # 16. Calculate the factorial of a number
        elif question_type == "Calculate the factorial of a number":
            num = random.randint(10, 100)
            question = f"Calculate the factorial of {num}."
            answer = math.factorial(num)
        # 17. Find the mode of a list of numbers
        elif question_type == "Find the mode of a list of numbers":
            array = [random.randint(113333, 113343) for _ in range(15)]
            question = f"Find the mode of the array {array}, give the
            ↪   result of multiplying it by 3."
            answer = max(set(array), key=array.count)
            answer = answer * 3
        # 18. Calculate the sum of even numbers in a list
        elif question_type == "Calculate the sum of even numbers in a
        ↪   list":
            array = [random.randint(1000, 1000000) for _ in
            ↪   range(random.randint(10, 25))]
```

```python
            question = f"Calculate the sum of even numbers in the list
            ↪    {array}."
            answer = sum(x for x in array if x % 2 == 0)
        # 19. Calculate the cumulative sum of an array
        elif question_type == "Calculate the cumulative sum of an array":
            array = [random.randint(1, 10000) for _ in
            ↪    range(random.randint(5, 15))]
            question = f"Calculate the cumulative sum of the array
            ↪    {array}."
            answer = [sum(array[:i+1]) for i in range(len(array))]
        # 20. Extract first N elements in a list
        elif question_type == "Extract first N elements in a list":
            N = random.randint(5, 10)
            array = [random.randint(1, 10000) for _ in
            ↪    range(random.randint(15, 35))]
            question = f"Extract first {N} elements in the list {array}
            ↪    and then plus 7 for each element in the sub-list."
            answer = array[:N]
            answer = [a + 7 for a in answer]
        # 21. Calculate cosine value
        elif question_type == "Calculate cosine value":
            degree = random.randint(0, 360) + 0.5
            question = f"Calculate cosine value for {degree} degree and
            ↪    round the result to two decimal places."
            answer = round(math.cos(math.radians(degree)), 2)
        # 22. Reverse the order of elements in a list
        elif question_type == "Reverse the order of elements in a list":
            array = [random.randint(1, 10000) for _ in
            ↪    range(random.randint(5, 15))]
            question = f"Reverse the order of the elements in the list
            ↪    {array} and then plus 3 for each element."
            answer = array[::-1]
            answer = [a + 3 for a in answer]
        # 23. Calculate the sum of squares of numbers in an array
        elif question_type == "Calculate the sum of squares of numbers in
        ↪    an array":
            array = [random.randint(10, 10000) for _ in
            ↪    range(random.randint(5, 15))]
            question = f"Calculate the sum of squares of the numbers in
            ↪    the array {array}."
            answer = sum(x ** 2 for x in array)
        # 24. Find the n-th smallest number in an array
        elif question_type == "Find the n-th smallest number in an
        ↪    array":
            array = [random.randint(1000, 10000000) for _ in
            ↪    range(random.randint(5, 15))]
            n = random.randint(1, len(array))
            question = f"Find the {n}-th smallest number in the array
            ↪    {array}, give the result of multiplying it by 3."
            answer = sorted(array)[n - 1] * 3
        # 25. Calculate the Euclidean distance between two points in a
        ↪    plane
        elif question_type == "Calculate the Euclidean distance between
        ↪    two points in a plane":
            x1, y1 = round(random.uniform(-100, 100), 2),
            ↪    round(random.uniform(-100, 100), 2)
            x2, y2 = round(random.uniform(-100, 100), 2),
            ↪    round(random.uniform(-100, 100), 2)
            question = f"Calculate the Euclidean distance between points
            ↪    ({x1}, {y1}) and ({x2}, {y2}), round the result to two
            ↪    decimal places."
            answer = round(math.sqrt((x2 - x1)**2 + (y2 - y1)**2), 2)
        # 26. Find the intersection of two strings
        elif question_type == "Find the intersection of two strings":
```

```python
            str1 = ''.join(random.choices('abcdefghijklmnopqrstuvwxyz',
            ↪  k=random.randint(50, 100)))
            str2 = ''.join(random.choices('abcdefghijklmnopqrstuvwxyz',
            ↪  k=random.randint(50, 100)))
            question = f"Find the intersection of string '{str1}' and
            ↪  string '{str2}'."
            answer = ''.join(set(str1) & set(str2))
        # 27. Calculate the compound interest given principal, rate, and
        ↪  time
        elif question_type == "Calculate the compound interest given
        ↪  principal, rate, and time":
            principal = random.randint(1000, 10000)
            rate = round(random.uniform(1, 10), 2)
            time = random.randint(1, 5)
            question = f"Calculate the compound interest for principal
            ↪  {principal}, rate {rate}%, and time {time} years, round
            ↪  the result to two decimal places."
            answer = round(principal * (1 + rate/100)**time, 2)
        # 28. Find the length of the longest word in a string
        elif question_type == "Find the length of the longest word in a
        ↪  string":
            words = [''.join(random.choices('abcdefghijklmnopqrstuvwxyz',
            ↪  k=random.randint(101, 200))) for _ in
            ↪  range(random.randint(5, 15))]
            string = ' '.join(words)
            question = f"Find the length of the longest word in the
            ↪  string '{string}'."
            answer = max(len(word) for word in words)
        # 29. Count the number of vowels in a string
        elif question_type == "Count the number of vowels in a string":
            string = ''.join(random.choices('abcdefghijklmnopqrstuvwxyz',
            ↪  k=random.randint(20, 50))) + 'a' * 101
            question = f"Count the number of vowels in the string
            ↪  '{string}'."
            answer = sum(1 for char in string if char in 'aeiou')
        # 30. Convert a list of Celsius temperatures to Fahrenheit
        elif question_type == "Convert a list of Celsius temperatures to
        ↪  Fahrenheit":
            celsius_list = [random.randint(-20, 40) for _ in range(5)]
            question = f"Convert the list of Celsius temperatures
            ↪  {celsius_list} to Fahrenheit, round the result to two
            ↪  decimal places."
            answer = [round(c * 9/5 + 32, 2) for c in celsius_list]
        # 31. Calculate time difference between two time zones
        elif question_type == "Calculate time difference between two time
        ↪  zones":
            tz1, tz2 = random.sample(pytz.all_timezones, 2)
            now = datetime.datetime.now()
            time1 = pytz.timezone(tz1).localize(now)
            time2 = pytz.timezone(tz2).localize(now)
            time_difference = abs((time1 - time2).total_seconds())
            question = f'Calculate time difference beween {tz1} and {tz2}
            ↪  in seconds.'
            answer = time_difference
        # 32. Find the leap year after a year
        elif question_type == "Find the leap year after a year":
            year = random.randint(1900, 2100)
            while calendar.isleap(year):
                year = random.randint(1900, 2100)
            question = f"Find the leap year after year {year}."
            answer = next(y for y in range(year + 1, year + 10000) if
            ↪  calendar.isleap(y))
        # 33. Find the most common word in a paragraph
        elif question_type == "Find the most common word in a paragraph":
```

```python
        words = ['apple', 'banana', 'orange', 'grape', 'pear',
        ↪  'hello', 'iphone', 'newspaper']
        paragraph = ' '.join(random.choices(words, k=30))
        question = f"Find the most common word in the paragraph
        ↪  '{paragraph}', concatenate it with the second common word
        ↪  in this paragraph."
        answer =
        ↪  Counter(paragraph.lower().split()).most_common(2)[0][0] +
        ↪  Counter(paragraph.lower().split()).most_common(2)[1][0]
    # 34. Calculate the perimeter of a rectangle given its length and
    ↪  width
    elif question_type == "Calculate the perimeter of a rectangle
    ↪  given its length and width":
        length, width = random.randint(100, 10000),
        ↪  random.randint(100, 10000)
        question = f"Calculate the perimeter of a rectangle with
        ↪  length {length} and width {width}."
        answer = 2 * (length + width)
    # 35. Sum all the digits of a given number
    elif question_type == "Sum all the digits of a given number":
        num = int(str(random.randint(100, 99999)) +
        ↪  '999999999999999')
        question = f"Sum all the digits of the number {num}."
        answer = sum(int(digit) for digit in str(num))
    # 36. Calculate the area of a triangle given its base and height
    elif question_type == "Calculate the area of a triangle given its
    ↪  base and height":
        base = round(random.uniform(100, 500), 2)
        height = round(random.uniform(100, 500), 2)
        question = f"Calculate the area of a triangle with base
        ↪  {base} and height {height}, round the result to two
        ↪  decimal places."
        answer = round(0.5 * base * height, 2)
    # 37. Find the real roots of a quadratic equation
    elif question_type == "Find the real roots of a quadratic
    ↪  equation":
        a = round(random.uniform(10, 200), 2)
        b = round(random.uniform(10, 200), 2)
        c = round(random.uniform(10, 200), 2)
        question = f"Find the real roots of the quadratic equation
        ↪  {a}x^2 + {b}x + {c} = 0, round the result to two decimal
        ↪  places."
        discriminant = b**2 - 4*a*c
        if discriminant > 0:
            root1 = (-b + math.sqrt(discriminant)) / (2*a)
            root2 = (-b - math.sqrt(discriminant)) / (2*a)
            answer = (round(root1, 2), round(root2, 2))
        elif discriminant == 0:
            root = -b / (2*a)
            answer = round(root, 2)
        else:
            answer = "no real roots"
    # 38. Calculate the sum of the cubes of a list
    elif question_type == "Calculate the sum of the cubes of a list":
        sequence = [random.randint(100, 10000) for _ in
        ↪  range(random.randint(5, 15))]
        question = f"Calculate the sum of the cubes of the list
        ↪  {sequence}."
        answer = sum([n**3 for n in sequence])
    # 39. Round all elements in a list to two decimal places
    elif question_type == "Round all elements in a list to two
    ↪  decimal places":
        array = [random.uniform(100, 10000) for _ in
        ↪  range(random.randint(5, 15))]
```

```python
            question = f"Round all elements in the list {array} to two
            ↪   decimal places."
            answer = [round(num, 2) for num in array]
        # 40. Find the first recurring word in a string
        elif question_type == "Find the first recurring word in a
        ↪   string":
            words = [''.join(random.choices('abcdefghijklmnopqrstuvwxyz',
            ↪   k=random.randint(5, 15))) for _ in
            ↪   range(random.randint(5, 10))]
            words = words * 3
            random.shuffle(words)
            paragraph = ' '.join(words)
            question = f"Find the first recurring word in the paragraph
            ↪   '{paragraph}', concatenate it with the second recurring
            ↪   word in this paragraph."
            def _find_recurring_words(paragraph):
                words = paragraph.lower().split()
                seen = set()
                first, second = None, None
                for word in words:
                    if word in seen:
                        if first is None:
                            first = word
                        elif second is None and word != first:
                            second = word
                            break
                    seen.add(word)
                return first + second
            answer = _find_recurring_words(paragraph)
        # 41. Calculate the hypotenuse of a right triangle given the
        ↪   other two sides
        elif question_type == "Calculate the hypotenuse of a right
        ↪   triangle given the other two sides":
            side1 = random.randint(100, 20000)
            side2 = random.randint(100, 20000)
            question = f"Calculate the hypotenuse of a right triangle
            ↪   with sides {side1} and {side2}, round the result to two
            ↪   decimal places."
            answer = round(math.sqrt(side1**2 + side2**2), 2)
        # 42. Extract all the numbers in a string
        elif question_type == "Extract all the numbers in a string":
            string1 = random.choices('abcdefghijklmnopqrstuvwxyz',
            ↪   k=random.randint(20, 50))
            string2 = random.choices('0123456789', k=random.randint(20,
            ↪   50))
            string = string1 + string2
            random.shuffle(string)
            string = ''.join(string)
            question = f"Extract all the numbers in the string '{string}'
            ↪   in order and concatenate them."
            answer = ''.join(re.findall(r'\d+', string))
        # 43. Convert a decimal number to its binary equivalent
        elif question_type == "Convert a decimal number to its binary
        ↪   equivalent":
            num = random.randint(1000, 1000000)
            question = f"Convert the decimal number {num} to its binary
            ↪   equivalent."
            answer = bin(num)[2:]
        # 44. Calculate the difference between two lists
        elif question_type == "Calculate the difference between two
        ↪   lists":
            list1 = [random.randint(1, 50) for _ in range(10)]
            list2 = [random.randint(1, 50) for _ in range(10)]
            question = f"Calculate the difference between the lists
            ↪   {list1} and {list2}."
```

```python
        answer = list(set(list1) - set(list2))
    # 45. Sum all odd numbers in a list
    elif question_type == "Sum all odd numbers in a list":
        array = [random.randint(1000, 1000000) for _ in
        ↪   range(random.randint(5, 15))]
        question = f"Sum all the odd numbers in the list {array}."
        answer = sum(x for x in array if x % 2 != 0)
    # 46. Find out all the numbers that are not unique
    elif question_type == "Find out all the numbers that are not
    ↪   unique":
        array = [random.randint(20, 35) for _ in range(20)]
        question = f"Find out all the numbers that are not unique in
        ↪   the array {array}."
        answer = [num for num, count in Counter(array).items() if
        ↪   count > 1]
    # 47. Flatten a 2D list into a 1D list
    elif question_type == "Flatten a 2D list into a 1D list":
        array_len = random.randint(2, 10)
        array = [[random.randint(1, 1000) for _ in range(array_len)]
        ↪   for _ in range(array_len)]
        question = f"Flatten the 2D list {array} into a 1D list."
        answer = [item for sublist in array for item in sublist]
    # 48. Remove duplicates from a list
    elif question_type == "Remove duplicates from a list":
        array = [random.randint(1, 20) for _ in range(15)]
        while len(array) == len(set(array)):
            array = [random.randint(1, 20) for _ in range(15)]
        question = f"Remove duplicates from the list {array}."
        answer = list(set(array))
    # 49. Generate the smallest N primes
    elif question_type == "Generate the smallest N primes":
        n = random.randint(5, 20)
        primes = []
        candidate = 2
        while len(primes) < n:
            if all(candidate % i != 0 for i in range(2, int(candidate
            ↪   ** 0.5) + 1)):
                primes.append(candidate)
            candidate += 1
        question = f"Generate the smallest {n} prime numbers."
        answer = primes
    # 50. Find the sum of all elements above the main diagonal of a
    ↪   matrix
    elif question_type == "Find the sum of all elements above the
    ↪   main diagonal of a matrix":
        matrix_len = random.randint(2, 10)
        matrix = [[random.randint(1000, 1000000) for _ in
        ↪   range(matrix_len)] for _ in range(matrix_len)]
        question = f"Find the sum of all elements above the main
        ↪   diagonal of the matrix {matrix}."
        answer = sum(matrix[i][j] for i in range(matrix_len) for j in
        ↪   range(i + 1, matrix_len))
    # 51. Filter elements in a list based on a condition
    elif question_type == "Filter elements in a list based on a
    ↪   condition":
        array = [random.randint(-100, 100) for _ in
        ↪   range(random.randint(10, 20))]
        condition = random.randint(-50, 50)
        question = f"Filter all elements in the array {array} that
        ↪   are greater than {condition}."
        answer = [x for x in array if x > condition]
    # 52. Merge two dictionaries into one
    elif question_type == "Merge two dictionaries into one":
        dict1 = {chr(65 + i): random.randint(1, 100) for i in
        ↪   range(random.randint(10, 20))}
```

```
436        dict2 = {chr(67 + i): random.randint(1, 100) for i in
         ↪  range(random.randint(10, 20))}
437        question = f"Merge the dictionaries {dict1} and {dict2},
         ↪  summing values for duplicate keys."
438        answer = {k: dict1.get(k, 0) + dict2.get(k, 0) for k in
         ↪  set(dict1) | set(dict2)}
439        # 53. Extract all words of a specific length from a text
440        elif question_type == "Extract all words of a specific length
         ↪  from a text":
441            text = '
             ↪  '.join([''.join(random.choices('abcdefghijklmnopqrstuvwxyz',
             ↪  k=random.randint(5, 10))) for _ in
             ↪  range(random.randint(10, 20))])
442            length = random.randint(5, 10)
443            question = f"Find all words in the text '{text}' that have
             ↪  exactly {length} characters."
444            answer = [word for word in text.split() if len(word) ==
             ↪  length]
445        # 54. Extract email addresses from a text
446        elif question_type == "Extract email addresses from a text":
447            answer = [Faker().email() for _ in range(random.randint(2,
             ↪  4))]
448            text = answer +
             ↪  [''.join(random.choices('abcdefghijklmnopqrstuvwxyz',
             ↪  k=random.randint(5, 10))) for _ in
             ↪  range(random.randint(10, 20))]
449            random.shuffle(text)
450            text = ' '.join(text)
451            question = f"Find all email addresses in the text: '{text}'"
452        # 55. Sort a list of strings by their length
453        elif question_type == "Sort a list of strings by their length":
454            strings =
             ↪  [''.join(random.choices('abcdefghijklmnopqrstuvwxyz',
             ↪  k=random.randint(5, 20))) for _ in
             ↪  range(random.randint(10, 20))]
455            question = f"Sort the list {strings} by the length of each
             ↪  string."
456            answer = sorted(strings, key=len)
457        # 56. Check if two strings are anagrams
458        elif question_type == "Check if two strings are anagrams":
459            string1 = random.choices('abcdefghijklmnopqrstuvwxyz',
             ↪  k=random.randint(10, 20))
460            string2 = random.choices('abcdefghijklmnopqrstuvwxyz',
             ↪  k=random.randint(10, 20)) if random.random() > 0.5 else
             ↪  string1
461            random.shuffle(string2)
462            string1 = ''.join(string1)
463            string2 = ''.join(string2)
464            question = f"Check if '{string1}' and '{string2}' are
             ↪  anagrams."
465            answer = sorted(string1) == sorted(string2)
466        # 57. Extract hashtags from a social media post
467        elif question_type == "Extract hashtags from a social media
         ↪  post":
468            topic = [''.join(random.choices('abcdefghijklmnopqrstuvwxyz',
             ↪  k=random.randint(5, 10))) for _ in
             ↪  range(random.randint(10, 20))]
469            hashtags = ['#' +
             ↪  ''.join(random.choices('abcdefghijklmnopqrstuvwxyz',
             ↪  k=random.randint(5, 10))) for _ in
             ↪  range(random.randint(2, 5))]
470            text = topic + hashtags
471            random.shuffle(text)
472            text = ' '.join(text)
473            question = f"Extract all hashtags from the post: '{text}'"
```

```
474        answer = [word for word in text.split() if
           ↪   word.startswith("#")]
475    # 58. Capitalize each word in a string
476    elif question_type == "Capitalize each word in a string":
477        text = '
           ↪   '.join([''.join(random.choices('abcdefghijklmnopqrstuvwxyz',
           ↪   k=random.randint(5, 10))) for _ in
           ↪   range(random.randint(10, 20))])
478        question = f"Capitalize each word in the string '{text}'."
479        answer = text.title()
480    # 59. Find the index of a substring in a string
481    elif question_type == "Find the index of a substring in a
       ↪   string":
482        string =
           ↪   [''.join(random.choices('abcdefghijklmnopqrstuvwxyz',
           ↪   k=random.randint(5, 10))) for _ in
           ↪   range(random.randint(10, 20))]
483        substring = random.choice(string)
484        string = ' '.join(string)
485        question = f"Find the index of the substring '{substring}' in
           ↪   the string '{string}'."
486        answer = string.find(substring)
487    # 60. Replace all vowels in a string with a specific character
488    elif question_type == "Replace all vowels in a string with a
       ↪   specific character":
489        string = '
           ↪   '.join([''.join(random.choices('abcdefghijklmnopqrstuvwxyz',
           ↪   k=random.randint(5, 10))) for _ in
           ↪   range(random.randint(10, 20))])
490        replacement = random.choice(["*", "$", "%", "&", "#", "@"])
491        question = f"Replace all vowels in the string '{string}' with
           ↪   '{replacement}'."
492        answer = ''.join([replacement if char.lower() in "aeiou" else
           ↪   char for char in string])
493    # not defined question
494    else:
495        raise ValueError(f'{question_type} is not defined')
496    # format and return
497    random_qa = {'question': question, 'answer': answer}
498    return random_qa
```

## A.8 PROMPTS FOR CONSTRUTING FACT

We construct the FACT datasets by prompting GPT-4o with,

```
1. Generate 100 Q&A pairs for LLM factual retrieval testing. The
↪  question topic should be related with Geography. Return them
↪  as a Python dictionary, with concise answers (3-5 words).
2. Generate 100 Q&A pairs for LLM factual retrieval testing. The
↪  question topic should be related with History. Return them as
↪  a Python dictionary, with concise answers (3-5 words).
3. Generate 100 Q&A pairs for LLM factual retrieval testing. The
↪  question topic should be related with Science. Return them as
↪  a Python dictionary, with concise answers (3-5 words).
4. Generate 100 Q&A pairs for LLM factual retrieval testing. The
↪  question topic should be related with Technology. Return them
↪  as a Python dictionary, with concise answers (3-5 words).
5. Generate 100 Q&A pairs for LLM factual retrieval testing. The
↪  question topic should be related with Mathematics. Return them
↪  as a Python dictionary, with concise answers (3-5 words).
6. Generate 100 Q&A pairs for LLM factual retrieval testing. The
↪  question topic should be related with Culture and Arts. Return
↪  them as a Python dictionary, with concise answers (3-5 words).
7. Generate 100 Q&A pairs for LLM factual retrieval testing. The
↪  question topic should be related with Sports. Return them as a
↪  Python dictionary, with concise answers (3-5 words).
8. Generate 100 Q&A pairs for LLM factual retrieval testing. The
↪  question topic should be related with Politics. Return them as
↪  a Python dictionary, with concise answers (3-5 words).
9. Generate 100 Q&A pairs for LLM factual retrieval testing. The
↪  question topic should be related with Language and Grammar.
↪  Return them as a Python dictionary, with concise answers (3-5
↪  words).
10. Generate 100 Q&A pairs for LLM factual retrieval testing. The
↪  question topic should be related with Current Affairs. Return
↪  them as a Python dictionary, with concise answers (3-5 words).
11. Generate 100 Q&A pairs for LLM factual retrieval testing. The
↪  question topic should be related with Entertainment. Return
↪  them as a Python dictionary, with concise answers (3-5 words).
12. Generate 100 Q&A pairs for LLM factual retrieval testing. The
↪  question topic should be related with Medicine and Health.
↪  Return them as a Python dictionary, with concise answers (3-5
↪  words).
13. Generate 100 Q&A pairs for LLM factual retrieval testing. The
↪  question topic should be related with Economics and Business.
↪  Return them as a Python dictionary, with concise answers (3-5
↪  words).
14. Generate 100 Q&A pairs for LLM factual retrieval testing. The
↪  question topic should be related with Religion and Mythology.
↪  Return them as a Python dictionary, with concise answers (3-5
↪  words).
15. Generate 100 Q&A pairs for LLM factual retrieval testing. The
↪  question topic should be related with General Knowledge.
↪  Return them as a Python dictionary, with concise answers (3-5
↪  words).
```

## A.9 SOME PROMISING RESULTS

Here, we show some promising results and compare them with other LLMs.

**Prompt:** *What is the 121st prime number?*
**Response [Llama3-8B SFT on ToolBridge]:** The 121st prime number is 661. \n\nTo find the 121st prime number, we can use the Sieve of Eratosthenes algorithm. This algorithm starts with a list of all the integers from 2 to 10000 (or any other upper limit), and then iterates through the list, marking as composite (not prime) all multiples of the first prime number. It then moves on to the next unmarked number, which is the next prime number, and repeats the process. This continues until all the numbers in the list have been marked as composite or prime. ✓
**Response [Llama3.1-70B-IT]:** The 121st prime number is 691. ✗
**Response [GPT-4, Web UI]:** The 121st prime number is 661. ✓
**Response [GPT-4o, Web UI]:** The 121st prime number is 661. ✓
**Response [Gemma2-27B-IT]:** The 121st prime number is 607. You can find prime numbers using primality tests or by looking them up in a list of prime numbers. ✗

**Prompt:** *What is the cosine of 20.4 degrees?*
**Response [Llama3-8B SFT on ToolBridge]:** The cosine of 20.4 degrees is approximately 0.9372819894918915. ✓
**Response [Llama3.1-70B-IT]:** The cosine of 20.4 degrees is approximately 0.9387. ✗
**Response [GPT-4, Web UI]:** The cosine of 20.4 degrees is approximately 0.9362. ✗
**Response [GPT-4o, Web UI]:** The cosine of 20.4 degrees is approximately 0.937. ✓
**Response [Gemma2-27B-IT]:** You'll need a calculator to find the cosine of 20.4 degrees. Here's how: Make sure your calculator is in degree mode. Enter "cos 20.4" and press enter. The result will be the cosine of 20.4 degrees. Let me know if you have any other questions. ✗

**Prompt:** *What is the area of a triangle with base 102.4 and height 215.9?*
**Response [Llama3-8B SFT on ToolBridge]:** The area of a triangle is given by the formula: A = (1/2) * base * height. \n\nIn this case, the base is 102.4 and the height is 215.9. \n\nSubstituting these values into the formula, we get: \n\nA = (1/2) * 102.4 * 215.9 = 11054.080000000002. ✓
**Response [Llama3.1-70B-IT]:** To find the area of a triangle, we use the formula: \n\nArea = (base × height) / 2\n\nGiven base = 102.4 and height = 215.9, we can plug in these values:\n\nArea = (102.4 × 215.9) / 2\n\nFirst, let's calculate the product of base and height:\n102.4 × 215.9 = 22123.36\n\nNow, divide the result by 2:\n22123.36 / 2 = 11061.68\n\nSo, the area of the triangle is approximately 11061.68 square units. ✗
**Response [GPT-4, Web UI]:** To find the area of a triangle, you can use the formula: Area = 1/2 × base × height. Plugging in the given values you provided: Area = 1/2 × 102.4 × 215.9 = 11041.92 square units. Therefore, the area of the triangle is 11041.92 square units. ✗
**Response [GPT-4o, Web UI]:** The area of a triangle is calculated using the formula: Area = 1/2 × base × height. Plugging in the given values: Area = 1/2 × 102.4 × 215.9 = 11053.12. So, the area of the triangle is 11,053.12 square units. ✗
**Response [Gemma2-27B-IT]:** The area of a triangle is calculated as (base * height) / 2. Therefore, the area of a triangle with base 102.4 and height 215.9 is: (102.4 * 215.9) / 2 = 11054.08. ✓

**Prompt:** *Which is the higher value of 9.9 or 9.11*
**Response [Llama3-8B SFT on ToolBridge]:** The higher value between 9.9 and 9.11 is 9.9. ✓
**Response [GPT-4o, Web UI]:** 9.9 is higher than 9.11 when considering their decimal values. This is because 9.9 is equivalent to 9.90, which is greater than 9.11. ✓
**Response [Gemini, Web UI]:** *9.11 is the higher value.* ✗

**Prompt:** *Find the GCD of 270 and 192.*
**Response [Llama3-8B SFT on ToolBridge]:** The GCD (Greatest Common Divisor) of 270 and 192 is 6. ✓
**Response [GPT-4o, Web UI]:** The GCD (Greatest Common Divisor) of 270 and 192 is: \n6. ✓
**Response [Gemini, Web UI]:** The GCD of 270 and 192 is 12. ✗

Figure 5: Some promising results from LLMs trained on ToolBridge.

A.10    EXTEND TO NATURAL WEB DATA

Our algorithm is not strictly reliant on the existing SFT datasets as it is equally applicable to natural data sourced from the web. Because, by using LLMs like GPT-4o, raw web data can be transformed into QA pairs, which can then be processed leveraging the proposed pipeline outlined in Section 3, including the selection, conversion, and filtering of valuable data entries.

For instance, Google's C4 dataset can be systematically transformed into QA pairs by using GPT-4o with the following example prompt:

```
You are an expert assistant tasked with creating concise and
↪  relevant Question-Answer (QA) pairs based on the content of a
↪  given web page. Follow these guidelines:

1. Content Focus: Read the provided web page content carefully and
↪  base all questions and answers solely on the information within
↪  the page. Do not include external knowledge.
2. Question Types: Include a mix of factual, explanatory, and
↪  inferential questions to test comprehension and understanding.
3. Question Structure: Ensure questions are clear and specific. Use
↪  diverse formats such as: What/Why/How questions.
4. Answer Structure: Provide direct, accurate, and concise answers.
↪  Avoid ambiguous or overly lengthy responses.

Example Web Content:

Coffee is one of the most popular beverages in the world. It is made
↪  from roasted coffee beans, which are seeds of the Coffea plant.
↪  A standard cup of coffee contains approximately 95 milligrams of
↪  caffeine. Studies suggest that consuming 400 milligrams of
↪  caffeine per day is generally safe for most adults. Many people
↪  drink coffee daily as part of their morning routine, with some
↪  consuming 2 to 4 cups per day depending on their preference.

Example Output:

- Question 1: How much caffeine is in a standard cup of coffee?
- Answer: A standard cup of coffee contains approximately 95
↪  milligrams of caffeine.
- Question 2: If a person drinks 3 cups of coffee, how much caffeine
↪  do they consume?
- Answer: They consume 285 milligrams of caffeine (95 * 3 = 285).
- Question 3: If a person drinks 2 cups of coffee in the morning and
↪  1 in the evening, how much caffeine do they consume in total?
- Answer: They consume 285 milligrams of caffeine (95 * 2 + 95 =
↪  285).
- Question 4: How many cups of coffee would a person need to drink
↪  to consume exactly 400 milligrams of caffeine?
- Answer: They would need to drink approximately 4.2 cups of coffee
↪  (400 ÷ 95 = 4.2).
- Question 5: If a person reduces their daily coffee intake from 4
↪  cups to 2 cups, how much less caffeine do they consume in a day?
- Answer: They consume 190 milligrams less caffeine (95 * 4 - 95 * 2
↪  = 380 - 190 = 190).

Web Content:

PLACEHOLDER

Output:
```

Some examples of the output results are presented below,

---

**Source data:**

Schwarz has been the president of the Greater Cleveland Film Commission for over ten years. He has over thirty years of experience in the film and television industries, working predominantly as a location manager and producer. His work includes "Wild At Heart," "My Name Is Earl," "Entourage," "The Comeback," "From the Earth to the Moon" and Tom Hanks and Steven Spielberg's "Band of Brothers," for which Ivan won a Golden Globe. The Film Commission is the only nonprofit organization dedicated to bringing jobs and economic impact to Northeast Ohio through the growth of a sustainable media production industry. Since passing the Ohio Motion Picture Tax Credit in 2009, more than 110 productions have shot in Cleveland, including "The Avengers," "Captain America: The Winter Soldier," and most recently, Matthew McConaughey's "White Boy Rick" and Helen Hunt's "I See You".

**Generated QA:**

- Question1: How long has Schwarz been the president of the Greater Cleveland Film Commission?
- Answer: Schwarz has been the president of the Greater Cleveland Film Commission for over ten years.
- Question2: What is the primary goal of the Greater Cleveland Film Commission?
- Answer: The primary goal of the Greater Cleveland Film Commission is to bring jobs and economic impact to Northeast Ohio through the growth of a sustainable media production industry.
- Question3: What significant legislation related to film production was passed in Ohio in 2009?
- Answer: The Ohio Motion Picture Tax Credit was passed in 2009.

**Source data:**

Earn monthly interest on our Citibank Time Deposits (also known as Fixed Deposits). What's more, you get to enjoy the flexibility of making partial withdrawals before maturity date of your Time Deposit. Partial withdrawals in multiples of RM5,000 before the maturity date. Option to pledge your Time Deposit for overdraft facility. Continue to earn the original interest rate with your remaining balance. View your Citibank® Time Deposit accounts across participating countries with a single login via Online Global View. Online account inquiries and banking transactions via Citibank® or Citigold® Online. Make banking inquiries anytime, anywhere via live e-Chat, 'Call Me' function and 20 seconds1 call pickup. Eligible for protection by PIDM**. * Monthly interest is only available for minimum deposit of RM50,000 and for a minimum tenure of 6 months. ** Click here for more details on PIDM protection. 1From the moment you press 0 to speak to a CitiPhone officer. For the Citibank Account Terms and Conditions, click here.

**Generated QA:**

- Question1: If a customer deposits RM100,000 into a Citibank Time Deposit account, how many partial withdrawals of RM5,000 can they make before the maturity date?
- Answer: They can make 20 partial withdrawals of RM5,000 each (RM100,000 ÷ RM5,000 = 20).
- Question2: A customer earns monthly interest on a Time Deposit of RM200,000 with an interest rate of 2% per annum. How much interest will they earn in one month?
- Answer: The monthly interest is RM333.33 (RM200,000 × 2% ÷ 12 = RM333.33).
- Question3: If a customer deposits RM75,000 for 6 months with an interest rate of 3% per annum, what is the total interest earned at the end of the tenure?
- Answer: The total interest earned is RM1,125 (RM75,000 × 3% ÷ 12 × 6 = RM1,125).
- Question4: A customer deposits RM120,000 for 12 months with a 4% annual interest rate. How much interest will they earn in total?
- Answer: The total interest earned is RM4,800 (RM120,000 × 4% = RM4,800).
- Question5: If the monthly interest on a Time Deposit of RM60,000 is RM100, what is the annual interest rate?
- Answer: The annual interest rate is 2% (RM100 × 12 ÷ RM60,000 = 2%).

---

Figure 6: Sample outcomes of converting natural web data from Google's C4 dataset into QA format with the help of GPT-4o.

## A.11 CONSTRUCT FACT WITH GEMINI

In Section 4.4, we employ GPT-4o, together with the prompts outlined in Appendix A.8, to construct the FACT dataset, which is utilized to evaluate the extent to which ToolBridge enhances the LLMs' capability for factual retrieval. Here, we also use Google Gemini to yield three batches of the FACT datasets termed Gemini-FACT-B(n), with each batch corresponding sequentially to the five prompts outlined in Appendix A.8. Table 12 demonstrates the evaluation results.

| Models | SFT data | Gemini-FACT-B1 | Gemini-FACT-B2 | Gemini-FACT-B3 |
|---|---|---|---|---|
| Llama3-8B | - | 75.8 | 52.5 | 60.3 |
| Llama3-8B-Lora | ToolBridge$^\S$ | 83.4 | 61.7 | 66.2 |
| Llama3-8B-Lora | ToolBridge | 89.2 | 63.3 | 71.2 |
| Mistral-7B | - | 77.5 | 59.2 | 67.8 |
| Mistral-7B-Lora | ToolBridge$^\S$ | 85.8 | 61.5 | 70.4 |
| Mistral-7B-Lora | ToolBridge | 90.8 | 64.7 | 74.7 |

Table 12: Experimental results on Gemini-FACT under zero-shot setting.

We can observe that the models trained on ToolBridge consistently achieve superior performance.

## A.12 DISTRIBUTION OF PYTHON PACKAGES ADOPTED IN TOOLBRIDGE

Here, we provide the distribution of all Python packages used in ToolBridge.

| Python Package | Frequency | Python Package | Frequency | Python Package | Frequency |
|---|---|---|---|---|---|
| math | 2669 | re | 2234 | sympy | 1838 |
| nltk | 1616 | datetime | 1512 | langdetect | 489 |
| numpy | 271 | collections | 248 | string | 246 |
| fractions | 209 | pandas | 205 | itertools | 121 |
| statistics | 118 | sklearn | 92 | io | 65 |
| requests | 54 | difflib | 35 | calendar | 34 |
| bs4 | 32 | os | 31 | random | 23 |
| functools | 23 | urllib | 20 | json | 17 |
| matplotlib | 12 | scipy | 10 | operator | 8 |
| xml | 8 | base64 | 7 | codecs | 6 |
| sys | 5 | PIL | 5 | bisect | 4 |
| csv | 3 | subprocess | 3 | cmath | 3 |
| time | 2 | pytz | 2 | ipaddress | 2 |
| decimal | 2 | unicodedata | 2 | dateutil | 2 |
| pytest | 2 | enum | 2 | heapq | 2 |
| keyword | 1 | typing | 1 | inspect | 1 |
| configparser | 1 | struct | 1 | ctypes | 1 |
| networkx | 1 | lxml | 1 | pylab | 1 |
| cycler | 1 | torch | 1 | html | 1 |
| textwrap | 1 | ast | 1 | locale | 1 |

Table 13: The usage frequency of all Python packages involved in ToolBridge.

A.13   REVIEW PROCESS FOR SFT DATASETS IN TABLE 1

The integration of Supervised Fine-tuning (SFT) datasets presented in Table 1 was primarily carried out by conducting keyword searches (*e.g.*, "supervised fine-tuning dataset", "SFT dataset", "instruction tuning dataset", and "LLM dataset") across platforms including Google Scholar, Hugging Face and GitHub. Representative search results include the following resources:

- https://github.com/Zjh-819/LLMDataHub,
- https://github.com/RenzeLou/awesome-instruction-learning,
- https://github.com/raunak-agarwal/instruction-datasets,
- https://github.com/zhilizju/Awesome-instruction-tuning,
- https://arxiv.org/abs/2402.18041,
- https://arxiv.org/abs/2402.06196.

Based on these resources, we conducted a manual review of all referenced datasets, including verifying whether the dataset qualified as an SFT dataset, assessing its open-source availability, identifying potential overlaps with existing collected datasets, and examining other potential concerns, such as copyright issues. At last, we derived Table 1.

## A.14 THE PROMPT SELECTION STRATEGY FOR LLAMA3-70B

To select an appropriate prompt for LLama3-70B to perform valuable data entries selection, we first designed the following candidate prompts:

```
Prompt1: Your task is to determine whether you can add calls to a
↪ Python API to a piece of text. The calls should help you get
↪ information required to complete the text. You only need to
↪ respond with "Yes" or "No", "Yes" means you can and "No" means
↪ you can't.

Input:

PLACEHOLDER

Output:

Prompt2: Determine if you can add Python API calls to the text to
↪ complete it. Respond with "Yes" or "No".

Input:

PLACEHOLDER

Output:

Prompt3: Prompt1 with example inputs outputs in Appendix A.1.

Prompt4: Prompt2 with example inputs outputs in Appendix A.1.
```

Then, we randomly sample 50 data entries from each dataset involved in Table 1 to serve as the test set for evaluating the effectiveness of these prompts. For each entry, five people label it to determine whether inserting Python code at the appropriate places can assist in yielding its subsequent contents. The final label for each data entry is determined based on the majority rule.

The evaluation results for each prompt are shown in the following table:

| Prompt ID | True Positive | False Positive | False Negative | True Negative | Recall | FPR |
|---|---|---|---|---|---|---|
| 1 | 781 | 452 | 0 | 17 | 100.0% | 96.4% |
| 2 | 781 | 432 | 0 | 37 | 100.0% | 92.1% |
| 3 | 732 | 10 | 49 | 459 | 93.7% | 2.1% |
| 4 | 679 | 25 | 102 | 444 | 86.9% | 5.3% |

Table 14: Ablation studies on candidate prompts for LLama3-70B.

Given the role of LLama3-70B as outlined in Section 3.2, the central aim at this stage is to maximize recall — safeguarding against the premature discarding of valuable data entries — while maintaining a sufficiently low false positive rate (FPR) to prevent excessive computational overhead for GPT-4o-mini in the subsequent stage.

In Section 3.3, we elaborate that GPT-4o-mini performs a secondary screening of the entries, refining and further filtering data entries deemed valueless. Its higher accuracy and generalizability compared to LLama3-70B make it the principal decision-making LLMs in our pipeline. However, owing to its significantly higher computational cost, it is essential to minimize the volume of data entries passed to GPT-4o-mini.

Based on the evaluation results in Table 14, Prompt 3 demonstrated the most balanced performance for our requirements, achieving high recall (93.7%) and a notably low FPR (2.1%). While Prompt 1 and Prompt 2 achieved perfect recall, their exceedingly high FPRs (96.4% and 92.1%, respectively)

made them unsuitable for the preliminary filtering task, as they would result in an excessive volume of data entries being forwarded to GPT-4o-mini. Conversely, Prompt 4's lower recall (86.9%) made it less effective at retaining valuable entries.

Considering the performance of Prompt 3, we determined it was sufficient to meet the requirements for LLama3-70B's role in the ToolBridge pipeline. Consequently, we selected Prompt 3 as the final prompt and did not conduct further ablation studies for LLama3-70B prompt design.

