# OpenReview forum: "ToolBridge: An Open-Source Dataset to Equip LLMs with External Tool Capabilities"
_ICLR.cc/2025/Conference — Submitted to ICLR 2025_

### Official Review · Reviewer_qAPt · 2024-10-30

**Soundness:** 3
**Presentation:** 2
**Contribution:** 3
**Rating:** 6
**Confidence:** 3

**Summary:**

This paper introduces ToolBridge, a collection of general open-source datasets designed to equip LLMs with effective external tool usage through a carefully structured pipeline. Experimental results demonstrate that fine-tuning on ToolBridge can enhance LLM performance across many benchmarks, including new benchmarks introduced in this paper for numerical calculation, factual retrieval, and data processing.

**Strengths:**

- this paper curates a collection of datasets through a carefully designed pipeline, and I really appreciate that they made it open source. I believe it can help with the research community
- I believe RandomQA will also be useful for further research

**Weaknesses:**

- The Figure 2 needs some revision, it's not informative, suggest including the processing details in each section
- Only llama models are used in experiments. Since llama is employed in the data curation pipeline, I suggest including other open or closed models with different sizes

**Questions:**

- What are the source datasets? Could you provide a brief description for each?
- I think RandomQA is a really good benchmark for numerical computation, but less data processing?
- Regarding the FACT, why do you prefer the current approach over using existing factual datasets and systematically processing them, for example, into a Python dictionary? I believe those datasets are typically more carefully constructed.

---

> ### Author Response · Authors · 2024-11-26
> **Response to Reviewer qAPt**
>
> We appreciate the reviewer’s thoughtful comments and suggestions. Below, we address each of the reviewer’s points in detail.
>
> **Comment 1: The Figure 2 needs some revision.**
>
> In the rebuttal revision, we have updated Figure 2 to include additional processing details for improved clarity and completeness. The revised figure now outlines the following key steps:
>
> - Collection and Reformatting of SFT Datasets.
> - Estimation and Ranking of Datasets.
> - Valuable data entries selection: Utilizing LLama3-70B to identify appropriate data entries.
> - Valuable data entries conversion: Leveraging GPT-4o-mini to insert Python code during the conversion of valuable data entries.
> - Valuable Data Entries Filtering: Performing consistency checks to filter and validate the final set of valuable data entries.
>
> These additions provide a more detailed and transparent depiction of the processing pipeline and thereby make Figure 2 more informative.
>
> **Comment 2: Only llama models are used in experiments.**
>
> In the rebuttal revision, we have also included the results of Mistral-7B in Tables 6-10. These results encompass:
>
> - Ablation studies on GSM 8k, GSM Plus and MathBench$^*$
>
> | Models|SFT Data|GSM 8k|GSM Plus|MathBench$^*$|
> |-|-|-|-|-|
> |Mistral-7B|-|38.1|25.1|27.8|
> |Mistral-7B-Lora|ToolBridge|45.0|29.8|31.0|
>
> - Ablation studies on Stanford WebQA
>
> | Models|SFT Data|Stanford WebQA|
> |-|-|-|
> |Mistral-7B|-|34.4|
> |Mistral-7B-Lora|ToolBridge|39.1|
>
> - Ablation studies on TruthfulQA
>
> | Models|SFT Data|ROUGE1|BLEURT|
> |-|-|-|-|
> |Mistral-7B|-|43.5|39.4|
> |Mistral-7B-Lora|ToolBridge|47.7|44.9|
>
> - Experimental results on RandomQA
>
> | Models|SFT Data|RandomQA-DP-B1|RandomQA-DP-B2|RandomQA-NC-B1|RandomQA-NC-B2|
> |-|-|-|-|-|-|
> |Mistral-7B|-|10.8|9.0|13.8|13.6|
> |Mistral-7B-Lora|ToolBridge|61.8|60.5|83.3|82.5|
>
> - Experimental results on FACT
>
> |Models|SFT Data|FACT-200-Batch1|FACT-200-Batch2|FACT-200-Batch3|
> |-|-|-|-|-|
> |Mistral-7B|-|85.0|67.5|65.9|
> |Mistral-7B-Lora|ToolBridge§|86.5|70.0|66.2|
> |Mistral-7B-Lora|ToolBridge|90.5|72.0|77.3|
>
> Collectively, these results provide strong evidence supporting the effectiveness of ToolBridge in various scenarios and tasks.
>
> **Comment 3: Source datasets and a brief description for each.**
>
> In Appendix A.6 of the original paper, we provided a detailed list of all source datasets along with their respective download links used in constructing ToolBridge. The rebuttal revision enhances this section by including concise descriptions for each dataset.
>
> **Comment 4: Less data processing templates in RandomQA.**
>
> In Appendix A.7 of the original paper, we provided 50 templates for constructing RandomQA, comprising 20 related to data processing and 30 related to numerical computation. To address the reviewer's comment regarding the imbalance, we have added 10 additional templates focusing on data processing in the rebuttal revision. The newly added templates include:
>
> - Filter elements in a list based on a condition,
> - Merge two dictionaries into one,
> - Extract all words of a specific length from a text,
> - Extract email addresses from a text,
> - Sort a list of strings by their length,
> - Check if two strings are anagrams,
> - Extract hashtags from a social media post,
> - Capitalize each word in a string,
> - Find the index of a substring in a string,
> - Replace all vowels in a string with a specific character.
>
> Furthermore, Table 9 has been updated to evaluate the performance of the data entries generated from the revised set of 60 templates, covering both data processing and numerical computation. The results of this evaluation are as follows:
>
> |Models|SFT Data|RandomQA-DP-B1|RandomQA-DP-B2|RandomQA-NC-B1|RandomQA-NC-B2|
> |-|-|-|-|-|-|
> |Llama2-7B|-|10.0|9.0|3.3|3.2|
> |Llama2-7B-Lora|ToolBridge§|19.2|16.6|7.7|8.6|
> |Llama2-7B-Lora|ToolBridge|53.2|54.0|63.4|60.7|
> |Llama3-8B|-|9.6|9.2|5.8|7.0|
> |Llama3-8B-Lora|ToolBridge§|30.3|29.0|15.8|13.9|
> |Llama3-8B-Lora|ToolBridge|62.1|60.0|82.1|80.1|
>
> Here, RandomQA-DP-B1/2 refers to data generated using templates related to data processing, while RandomQA-NC-B1/2 corresponds to data generated using templates related to numerical computation. These results demonstrate that LLMs effectively learn to leverage external tools, thereby enhancing their capabilities in fundamental tasks such as data processing and numerical computation when trained on ToolBridge.
>
> **Comment 5: Evaluate on existing factual datasets**
>
> In response to the reviewer's comment, we have included Table 8 in the rebuttal revision to evaluate the model's performance on TruthfulQA, a widely used factual dataset. The evaluation results are as follows:
>
> |Models|SFT Data|ROUGE1|BLEURT|
> |-|-|-|-|
> |Llama3-8B|-|41.2|34.6|
> |Llama3-8B-Lora|ToolBridge§|47.0|42.8|
> |Llama3-8B-Lora|ToolBridge|48.7|44.4|
>
> These results demonstrate that ToolBridge can enhance the factual retrieval capabilities of LLMs, as reflected in the improved ROUGE1 and BLEURT scores.

---

> ### Comment · Reviewer_qAPt · 2024-12-02
>
> Thank you so much for addressing my questions. I believe the score appropriately reflects the merits of the work.

---

### Official Review · Reviewer_NXYV · 2024-10-30

**Soundness:** 2
**Presentation:** 4
**Contribution:** 2
**Rating:** 5
**Confidence:** 4

**Summary:**

This paper introduces ToolBridge, a dataset intended to enhance large language models' (LLMs) capacity for effectively employing external tools, and addressing gaps in model training. ToolBridge integrates various open-source SFT data, compiling an extensive dataset to train LLMs for some tasks that can be enhanced by using Python. The authors highlight the need for high-quality, transparent datasets specifically designed for tool integration and propose that they will release the ToolBridge dataset and data process pipeline.

**Strengths:**

1. The paper describes a pipeline for generating tool-using data and presents a dataset (though direct access to the dataset is not provided here) to address gaps in large language model tool-usage capabilities.
2. The structured data-filtering approach introduced in this paper effectively enhances dataset quality and improves the ability of tool using(function calling).
3. The authors report that, following training on the ToolBridge dataset, the model shows improved performance in mathematical, fact and RandomQA benchmarks.

**Weaknesses:**

1. This paper restricts external tool integration to Python APIs. Although the dataset includes 63 Python packages, this selection may be limited compared to other tool-using datasets, such as ToolBench.
2. A comparison with other tool-using datasets, like ToolBench, may further clarify ToolBridge's distinct contributions.
3. The paper could benefit from a more detailed explanation of the fact-checking process. While it states that the model can request external web pages using request packages, it remains unclear whether the URLs are generated by the model, predefined, or retrieved through other methods.
4. This paper does not evaluate the factual retrieval abilities of LLM against established benchmarks but uses a self-construct dataset. There are some existing benchmarks, such as TruthfulQA or HaluEval, to assess the model's accuracy and reliability in factual responses, which may be more convincing.

**Questions:**

See weakness.

---

> ### Author Response · Authors · 2024-11-26
> **Response to Reviewer NXYV**
>
> We sincerely thank the reviewer for the insightful comments and constructive suggestions. Below, we have provided detailed responses to each point raised in the review.
>
> **Comment1: Comparison with Function Calling Datasets such as ToolBench**
>
> We would like to clarify that the dataset proposed in this paper is designed specifically for **tool former** approaches, which address a distinct problem space compared to **function calling**. These two paradigms serve different objectives and employ fundamentally different methodologies:
>
> - **Function calling** focuses on enabling LLMs to act as agents that invoke APIs without requiring explicit training on their usage. This is achieved through curated datasets and standardized evaluation protocols that assess and compare the tool-use capabilities of LLMs. Function calling emphasizes streamlined API integration to perform predefined tasks effectively.
>
> - **Tool former**, on the other hand, seeks to empower LLMs with the capability to autonomously invoke and utilize external tools during inference. This approach emphasizes advanced reasoning, enabling LLMs to independently determine which external tool to call, the optimal timing for the call, and how to parse and integrate the execution results systematically into their generated responses.
>
> Given these differences, ToolBridge is not directly comparable to function calling datasets such as ToolBench, as their objectives and methodologies differ significantly.
>
> The primary contribution of this paper, to the best of our knowledge, is the first open-sourcing of training data specifically tailored for the tool former domain. This dataset enables LLMs to learn how to autonomously and effectively utilize external tools. We have explicitly addressed this distinction in the rebuttal revision, particularly in lines 40–49 of the Introduction section and lines 129–161 of the Related Work section, to better highlight our contributions and mitigate potential misunderstandings.
>
> **Comment2: Only 63 Python packages included in ToolBridge**
>
> First, this paper focuses on tool former, not function calling, and thus a direct comparison with ToolBench is not appropriate or meaningful. These two approaches target fundamentally different objectives, as outlined in our response to Comment 1.
>
> Second, the number of Python packages included in ToolBridge is not directly indicative of the dataset's breadth or functionality. A single Python package can encompass a wide range of functions with diverse capabilities. To provide a clearer perspective on the dataset’s richness, we conducted a detailed analysis and found that ToolBridge includes a total of 125,651 distinct function calls. This reflects the dataset's considerable scale and functional diversity.
>
> **Comment3: More Detailed Explanation of the Fact-checking Process**
>
> The detailed inference process of LLMs trained on ToolBridge is outlined in Algorithm 1 of the original paper. Briefly, ToolBridge enables the model to autonomously generate external link addresses required for retrieving factual information.
>
> For example, given the query, "What county is Farmington Hills, MI in?", the model generates the following Python code:
>
> ```python
> import requests
> from bs4 import BeautifulSoup
>
> response = requests.get('https://en.wikipedia.org/wiki/Farmington_Hills,_Michigan')
> soup = BeautifulSoup(response.content, 'html.parser')
> county = soup.find('th', text='County').find_next_sibling('td').text.strip()
>
> print(county)
> ```
>
> Upon detecting special tokens <python> and </python>, the Python interpreter executes the code. The requests and BeautifulSoup libraries retrieve and parse the webpage to extract the county name, which is then returned within <result> tags (*i.e.*, <result>Oakland</result>). This result informs subsequent text generation, demonstrating how ToolBridge enhances factual accuracy by leveraging external tools.
>
> **Comment4: Evaluate the factual retrieval abilities of LLM against established benchmarks**
>
> We appreciate the reviewer’s suggestion and have included an evaluation of the model's factual retrieval abilities using the TruthfulQA benchmark in the rebuttal revision (Table 8). The results are summarized below:
>
> |Models|SFT Data|ROUGE1|BLEURT|
> |-|-|-|-|
> |Llama3-8B|-|41.2|34.6|
> |Llama3-8B-Lora|ToolBridge§|47.0|42.8|
> |Llama3-8B-Lora|ToolBridge|48.7|44.4|
> |Mistral-7B|-|43.5|39.4|
> |Mistral-7B-Lora|ToolBridge§|44.9|42.3|
> |Mistral-7B-Lora|ToolBridge|47.7|44.9|
>
> The results demonstrate that ToolBridge can improve the factual retrieval capabilities of LLMs, as reflected in the higher ROUGE-1 and BLEURT scores compared to baseline models.

---

### Official Review · Reviewer_5jrH · 2024-11-04

**Soundness:** 3
**Presentation:** 2
**Contribution:** 3
**Rating:** 6
**Confidence:** 3

**Summary:**

This paper introduces ToolBridge, a new dataset for finetuning models to run Python code to support the completion of data processing, numerical computation, and factual retrieval tasks. To construct this dataset, the authors begin with a pool of 13 supervised finetuning datasets. Then, they use Llama 3 70B to filter for dataset examples that can make use of external tool use, and use GPT-4o-mini to create the Python code that invokes external tools. They show that by finetuning on ToolBridge, models can improve task performance.

**Strengths:**

The authors do a good job of motivating the purpose of their contribution, which emphasizes the open-source nature of their work juxtaposed against existing work.

**Weaknesses:**

Some details for assessing the quality of this dataset and the quality of their data curation pipeline are missing. For instance, how well does Llama3-70B select “valuable” data entries (3.2) (e.g. false positive, false negative rates)? Did you experiment with other prompts for selecting data, and if so, how did you decide on your final prompt? Generally, it would be good to get some sense of what it means for a data entry to be “valuable” or suitable for tool use, and how you selected the few-shot examples in the prompts you provide.

The authors’ data and experiments are essentially a case study of model distillation. That is, they use potentially stronger models (Llama-3-70B, GPT-4o-mini) to create data for weaker ones (Llama-3-8B, Llama-2-7B). It would be more interesting to see how data generated by strong models can improve their own performance.

The authors could also provide more information on the composition of their dataset. They provide some examples of Python packages involved, but which ones are most common and what does the distribution look like? Approximately how many dataset examples pertain to the three key tasks of numerical calculations, factual retrieval, and data processing? It may also be useful to disaggregate RandomQA, which the authors create themselves, based on task type to show how ToolBridge may boost performance for some tasks more so than others.

The qualitative results section in this paper is pretty lacking. It essentially just shows three examples of prompts to which Llama3-8B trained on ToolBridge answers correctly, along with other model’s responses (which are sometimes correct and sometimes not). There is no qualitative analysis present in this section; it’s just three input/output examples. These examples don’t really add much to the paper, and the models included in Figure 3 also seem a bit random, because these models are not mentioned at all among the paper’s quantitative results. To revise, I suggest that the authors do one of the following:
- Actually conduct some qualitative analysis of results. For example, the authors could characterize what kinds of examples Llama3-8B trained on ToolBridge does best on and which ones it tends to struggle on (like the bullet list of deficiencies outlined in lines 472-481). To do this, I suggest the authors look up how to do “qualitative coding”.
- Do not include a “qualitative results” section at all, and instead replace Figure 3 with a figure of example inputs/outputs from the four models experimented with in 4.3 and/or 4.4.
- Keep Figure 3, but also include quantitative results in Tables 6-9 for Llama3.1-70B-IT, GPT-4, GPT-4o, and Gemma2-27B-IT.

More minor comments:
- Line 187: “Llama3” -> “Llama3-70B”
- Line 208: “using the Llama3-70B” -> “using Llama3-70B”
- Line 359: “four baseline models: the base model of Llama2-7B and Llama3-8B” This says “four” but names two models.
- Line 484: “llama3-8B” -> “Llama3-8B” (to match model naming elsewhere in the paper)

**Questions:**

Why do you run Llama 3 70B filtering twice (once in line 187, and once in line 206)? Why not just run it once over all datasets?

How did you conduct your review of SFT datasets to finalize the ones you include in Table 1? That is, how did you ensure that it was sufficiently “comprehensive” as you say?

For the random sampling done in line 187, how many examples did you sample to make $\mathcal{S}_i$ for each dataset?

Line 196: What is $N$ and how did you determine what this value would be for each dataset?

Are there any substantive differences between the three different batches of the FACT benchmark? That is, why is performance reported for each batch instead of aggregating performance across them?

---

> ### Author Response · Authors · 2024-11-27
> **Response to Reviewer 5jrH - Part 1**
>
> We greatly value the reviewer’s thoughtful comments and helpful suggestions. Below, we offer detailed responses to each of the concerns raised.
>
> **Comment1: Run Llama 3 70B filtering twice**
>
> We employed a two-stage filtering process with Llama 3 70B to balance computational efficiency and maximize the retrieval of valuable data. In the first stage, a subset of each dataset is sampled to estimate the proportion of valuable data (*i.e.*, suitable for invoking external tools to help LLMs’ reasoning process) within that dataset. This preliminary assessment allows us to rank datasets based on their relative quality. In the second stage, we conduct a full-scale filtering process only on the datasets identified as having a high proportion of valuable data from the first stage.
>
> This staged approach addresses the significant computational challenges associated with the sheer volume of candidate datasets provided by the community. Running the Llama 3 70B filtering process directly on all data across all datasets would be prohibitively time-consuming. By focusing the second round of filtering on high-quality datasets, we achieve significant efficiency gains. Specifically, for the current scale of data, this approach reduces the total inference time by approximately 40%.
>
> **Comment2: Review Process of SFT Datasets**
>
> The integration of Supervised Fine-tuning (SFT) datasets presented in Table 1 was primarily carried out by conducting keyword searches (*e.g.*, "supervised fine-tuning dataset", "SFT dataset", "instruction tuning dataset", and "LLM dataset") across platforms including Google Scholar, Hugging Face and GitHub.
> Representative search results include the following resources:
>
> - https://github.com/Zjh-819/LLMDataHub,
> - https://github.com/RenzeLou/awesome-instruction-learning,
> - https://github.com/raunak-agarwal/instruction-datasets,
> - https://github.com/zhilizju/Awesome-instruction-tuning,
> - https://arxiv.org/abs/2402.18041,
> - https://arxiv.org/abs/2402.06196.
>
> Based on these resources, we conducted a manual review of all referenced datasets, including verifying whether the dataset qualified as an SFT dataset, assessing its open-source availability, identifying potential overlaps with existing collected datasets, and examining other potential concerns, such as copyright issues.
> At last, we derived Table 1.
>
> To enhance transparency and reproducibility, we have included a detailed explanation of this process in Appendix A.13 of the rebuttal revision. Additionally, we have revised the phrasing in the manuscript to replace "a comprehensive review" with "a review" to prevent potential misinterpretation regarding our study.
>
> **Comment3: Sampling ratio for obtaining $\mathcal{S}_i$ in Section 3.1**
>
> To obtain $\mathcal{S}_i$, we sampled 1% of examples from $\mathcal{D}_i$ for each dataset. We have clarified this point in the rebuttal revision (line 200) to enhance transparency and ensure reproducibility of our methodology.
>
> **Comment4: Selection of $N$ in Section 3.1**
>
> By default, we randomly sample $N = 100$ data entries from $\mathcal{D}_i$ for manual review to identify the presence of irrelevant characters or content. This number has been clarified in the rebuttal revision (line 210) to enhance the transparency of our pipeline. The choice of $N$ is informed by two key considerations: (1) the cost of manual review, as excessively large values of $N$ would impose a substantial workload; and (2) the observation that datasets constructed through methods such as web crawling typically have a high likelihood of including irrelevant content. Based on our empirical experience, we find that using a relatively small $N$ introduces minimal impact on the overall results, as larger values do not significantly alter the identification of irrelevant entries.
>
> **Comment5: Different batches of FACT**
>
> The data for the three batches of FACT were generated using GPT-4o with distinct prompts. Specifically, the first five prompts listed in Appendix A.8 were utilized to construct Batch 1, the next five prompts were used for Batch 2, and the final five prompts were employed for Batch 3. This process has been explicitly clarified in lines 513 to 519 of the rebuttal revision to ensure transparency and to avoid any potential confusion regarding different batches of FACT.
>
> **Comment6: Typo Issues**
>
> We have addressed and corrected the typo issues identified in lines 187, 208, 359, and 484 of the original paper. These corrections have been incorporated into the rebuttal revision to ensure clarity and accuracy in the revised manuscript.
>
> **Comment7: Qualitative Results Section**
>
> We appreciate your insightful suggestion. In the rebuttal revision, we have removed Section 3.5 from the main text and relocated the example inputs and outputs to Appendix A.9 to improve the manuscript.

---

> > ### Author Response · Authors · 2024-11-27
> > **Response to Reviewer 5jrH - Part 2**
> >
> > **Comment8: Distribution of Python Packages in ToolBridge**
> >
> > Thank you for your suggestion. We have provided a detailed frequency distribution of the Python packages used in ToolBridge in Appendix A.12. Below is the specific distribution:
> >
> > |Python Package|Frequency|Python Package|Frequency|Python Package|Frequency|Python Package|Frequency|
> > |-|-|-|-|-|-|-|-|
> > |math|2669|re|2234|sympy|1838|langdetect|489|
> > |nltk|1616|datetime|1512|collections|248|string|246|
> > |numpy|271|pandas|205|itertools|121|sklearn|92|
> > |fractions|209|diffllib|35|calendar|34|os|31|
> > |statistics|118|bs4|32|random|23|functools|23|
> > |requests|54|urllib|20|json|17|scipy|10|
> > |matplotlib|12|xml|8|operator|8|base64|7|
> > |sys|5|PIL|5|codecs|6|bisect|4|
> > |csv|3|subprocess|3|cmath|3|pytz|2|
> > |time|2|ipaddress|2|decimal|2|dateutil|2|
> > |pytest|2|unicodedata|2|enum|2|heapq|2|
> > |keyword|1|typing|1|struct|1|lxml|1|
> > |configparser|1|ctypes|1|networkx|1|pylab|1|
> > |cycler|1|html|1|textwrap|1|locale|1|
> >
> > **Comment9: Disaggregate RandomQA**
> >
> > In the rebuttal revision, we disaggregated RandomQA into two distinct components—Data Processing (DP) and Numerical Computation (NC)—to conduct a more granular evaluation. This division allows us to better analyze how ToolBridge contributes to the model's performance in these two fundamental aspects independently. The detailed results are summarized below:
> >
> > |Models|SFT Data|RandomQA-DP-B1|RandomQA-DP-B2|RandomQA-NC-B1|RandomQA-NC-B2|
> > |-|-|-|-|-|-|
> > |Llama2-7B|-|10.0|9.0|3.3|3.2|
> > |Llama2-7B-Lora|ToolBridge§|19.2|16.6|7.7|8.6|
> > |Llama2-7B-Lora|ToolBridge|53.2|54.0|63.4|60.7|
> > |Llama3-8B|-|9.6|9.2|5.8|7.0|
> > |Llama3-8B-Lora|ToolBridge§|30.3|29.0|15.8|13.9|
> > |Llama3-8B-Lora|ToolBridge|62.1|60.0|82.1|80.1|
> >
> > These results clearly demonstrate that ToolBridge enables LLMs to effectively utilize external tools, significantly improving their performance in both data processing and numerical computation tasks.
> >
> > **Comment10: A Case Study of Model Distillation**
> >
> > We would like to clarify that our approach is fundamentally different from model distillation, as explained below:
> >
> > - **Data Processing and Filtering**. In Sections 3.3 and 3.4, our pipeline does not retain all data entries processed by GPT-4o-mini as ToolBridge entries. Instead, we apply an algorithmic filtering and processing mechanism that discards erroneous or low-quality entries generated by GPT-4o-mini, reducing the dataset size significantly from 1,527,153 entries to 178,023 entries. This process is distinctly different from model distillation, which typically involves transferring knowledge from a teacher model to a student model **in its entirety**.
> >
> > - **Role of LLama3-70B in the Pipeline**. LLama3-70B is utilized exclusively for data filtering, not for generating the data itself. The data used in ToolBridge originates from the open-source community and processed by GPT-4o-mini. Thus, LLama3-70B’s role does not align with the principles of model distillation, where a smaller model learns directly from the outputs of a larger model.
> >
> > - **Focus on Tool-Former Paradigm**. ToolBridge focuses on tool-former paradigms rather than simple function-calling paradigms. Thus,  LLMs are trained not merely to yield code for invoking external tools but to holistically learn which tools to invoke, identify the optimal timing for these invocations, and systematically parse and integrate the execution results into their responses. However, GPT-4o-mini is specifically utilized to identify the precise locations within each entry where external tools should be invoked and to yield the corresponding tool-invocation code, rather than synthesizing complete entries directly. This also underscores the difference between our approach and model distillation.
> >
> > To further validate the effectiveness of ToolBridge and differentiate it from model distillation, we have conducted ablation studies with Llama3-70B. These results are included in Tables 6, 9, and 10 of the revised manuscript. Below, we summarize the results:
> >
> > - Ablation studies on GSM 8k, GSM Plus and MathBench$^*$
> >
> > |Models|SFT Data|GSM 8k|GSM Plus|MathBench$^*$|
> > |-|-|-|-|-|
> > |Llama3-70B|-|75.3|54.4|42.1|
> > |Llama3-70B-Lora|ToolBridge§|78.5|57.6|44.1|
> > |Llama3-70B-Lora|ToolBridge|80.1|59.8|46.9|
> >
> > -  Experimental results on RandomQA
> >
> > |Models|SFT Data|RandomQA-DP-B1|RandomQA-DP-B2|RandomQA-NC-B1|RandomQA-NC-B2|
> > |-|-|-|-|-|-|
> > |Llama3-70B|-|20.0|17.1|9.6|8.9|
> > |Llama3-70B-Lora|ToolBridge§|32.1|31.7|22.0|20.3|
> > |Llama3-70B-Lora|ToolBridge|74.2|69.9|89.7|89.1|
> >
> > -  Experimental results on FACT
> >
> > |Models|SFT Data|FACT-200-Batch1|FACT-200-Batch2|FACT-200-Batch3|
> > |-|-|-|-|-|
> > |Llama3-70B|-|76.0|53.5|54.0|
> > |Llama3-70B-Lora|ToolBridge§|88.3|72.4|70.7|
> > |Llama3-70B-Lora|ToolBridge|91.2|74.6|82.6|
> >
> > These results consistently demonstrate the superior performance of ToolBridge, underscoring its distinctiveness from model distillation.

---

> > > ### Author Response · Authors · 2024-11-27
> > > **Response to Reviewer 5jrH - Part 3**
> > >
> > > **Comment11: The Prompt Selection Strategy For LLama3-70B**
> > >
> > > In developing ToolBridge, we explored the design of prompts used by LLama3-70B to ensure good performance in its role of preliminary data filtering. During the initial exploration, we formulated four candidate prompts based on our research experience:
> > >
> > > - Prompt 1: '''Your task is to determine whether you can add calls to a Python API to a piece of text. The calls should help you get information required to complete the text. You only need to respond with "Yes" or "No", "Yes" means you can and "No" means you can't. \n Input:\nPLACEHOLDER\nOutput:\n'''
> > >
> > > - Prompt 2: '''Determine if you can add Python API calls to the text to complete it. Respond with "Yes" or "No". \n Input:\nPLACEHOLDER\nOutput:\n'''
> > >
> > > - Prompt 3: Prompt 1 with example inputs outputs described in Appendix A.1.
> > >
> > > - Prompt 4: Prompt 2 with example inputs outputs described in Appendix A.1.
> > >
> > > For Prompts 3 and 4, the few-shot examples were generated based on suggestions from GPT-4o, offering illustrative guidance for LLama3-70B. To evaluate these prompts, we constructed a test set by randomly sampling 50 entries from each dataset listed in Table 1. Five annotators independently labeled each entry based on whether Python API insertion could assist in completing the text. Final labels were determined via majority voting.
> > >
> > > The evaluation results are summarized in the table below:
> > >
> > > |Prompt ID|True Positive|False Positive|False Negative|True Negative|Recall|FPR|
> > > |-|-|-|-|-|-|-|
> > > |1|781|452|0|17|100.0%|96.4%|
> > > |2|781|432|0|37|100.0%|92.1%|
> > > |3|732|10|49|459|93.7%|2.1%|
> > > |4|679|25|102|444|86.9%|5.3%|
> > >
> > > Given the role of LLama3-70B as described in Section 3.2, the primary objective at this stage is to maximize recall—ensuring that valuable data entries are not prematurely discarded—while maintaining a sufficiently low false positive rate (FPR) to prevent excessive computational overhead for GPT-4o-mini in the subsequent stage.
> > >
> > > In Section 3.3, we elaborate that GPT-4o-mini performs a secondary screening of the data, refining and further filtering entries deemed valueless. Its higher accuracy and generalizability compared to LLama3-70B make it the principal decision-making model in our pipeline. However, due to its significantly higher computational cost, it is crucial to minimize the volume of data passed to GPT-4o-mini without compromising the inclusion of potentially valuable entries.
> > >
> > > Based on the evaluation results, Prompt 3 demonstrated the most balanced performance for our requirements, achieving high recall (93.7%) and a notably low FPR (2.1%). While Prompt 1 and Prompt 2 achieved perfect recall, their exceedingly high FPRs (96.4% and 92.1%, respectively) rendered them unsuitable for the preliminary filtering task, as they would lead to excessive data being passed to GPT-4o-mini. Conversely, Prompt 4's lower recall (86.9%) made it less effective at retaining valuable entries.
> > >
> > > Considering the performance of Prompt 3, we determined it was sufficient to meet the requirements for LLama3-70B's role in the ToolBridge pipeline. Consequently, we selected Prompt 3 as the final prompt and did not conduct further ablation studies for LLama3-70B prompt design.
> > >
> > > The aforementioned content has been incorporated into Appendix A.14 of the rebuttal revision to enhance the transparency of the pipeline employed in constructing ToolBridge.

---

### Official Review · Reviewer_X8CW · 2024-11-04

**Soundness:** 2
**Presentation:** 3
**Contribution:** 2
**Rating:** 5
**Confidence:** 3

**Summary:**

The paper presents a novel post-training dataset, ToolBridge, aimed at enhancing the external tool usage capabilities of LLMs.

1. The authors consolidate a broad set of open-access datasets to form a comprehensive dataset pool. This pool undergoes a systematic process that includes sampling, code generation, and consistency validation to identify entries suitable for tool invocation.

2. The ToolBridge pipeline features three key phases—selection, conversion, and filtering. Entries are reformatted, enriched with Python code snippets by GPT-4 for tool calls, and subsequently validated to ensure the consistency of execution results with the corresponding responses.

3. The authors conducted experiments using both standard math and QA benchmarks as well as custom benchmarks. Results show consistent improvements over the base model after SFT with ToolBridge.

**Strengths:**

1. The authors effectively assemble a large and diverse collection of existing datasets. Drawing from various sources, they create a robust dataset pool that enhances the relevance and generalizability of ToolBridge for training language models.

2. The pipeline for data conversion, validation, and tool invocation is meticulously designed, reflecting significant effort. The authors provide transparency in each step of the process, from dataset sampling to code generation and consistency checks, making it easier for the community to use.

**Weaknesses:**

1. The authors rely entirely on ensembling existing datasets, limiting their contribution's novelty and scope. This also makes the pipeline depend on pre-existing collections of instruction following/post-training data, and cannot be derived from natural data on the web.

2. The pipeline primarily modifies existing data by inserting Python-based tool calls. it should consider including stronger baseline models that focus more on coding tasks, such as Code Llama.

3. The evaluation raises questions, particularly in relation to benchmark selection. Since the dataset is targeted at tool use, it would be more appropriate to evaluate with dedicated tool-use benchmarks. Additionally, the performance gaps between ToolBridge and ToolBridge* (a variant without tool calls) on standard benchmarks are small, which brings concern of how much extra value is added through the pipeline rather than the underlying SFT data from existing works.

4. While the model shows greater improvement on custom datasets, these evaluations raise concerns. The RandomQA dataset is mostly template-generated, and the FACT dataset is generated by GPT-4. Given that much of ToolBridge data also originates from GPT-4, this could introduce potential biases.

**Questions:**

1. Why not use more coding-focused base models?
2. Is it possible to include some dedicated tool use benchmarks?
3. Why use WebQA and FACT for evaluation? Is the purpose to test the ability to use search tools?
4. Is all methods evaluated use the same prompt and tool use support? Since the baselines are not tuned for the specific format, how often do they use the tools, or write code. Some more detail on the experiment setup could help understand the improvement on tool_use as separated from just encouraging tool use under the specific format.

---

> ### Author Response · Authors · 2024-11-26
> **Response to Reviewer X8CW - Part 1**
>
> We are grateful to the reviewer for the valuable and insightful feedback. Below, we have provided a thorough and detailed response to each of the comments and suggestions raised.
>
> **Comment1: Relies on Existing SFT Datasets**
>
> We would like to clarify that the algorithm for constructing ToolBridge is a flexible and adaptable pipeline. This pipeline is capable of processing raw web data by transforming the data into structured QA pairs with the assistance of LLMs like GPT-4o. Here is an example of QA pairs obtained by using GPT-4o to transform the raw web data from the publicly available C4 dataset,
>
> - Question: If a customer deposits RM100,000 into a Citibank Time Deposit account, how many partial withdrawals of RM5,000 can they make before the maturity date?
> - Answer: They can make 20 partial withdrawals of RM5,000 each (RM100,000 ÷ RM5,000 = 20).
>
> Through this transformation, the pipeline enables the subsequent selection, conversion, and filtering of data entries, ultimately ensuring the generation of high-quality entries for the ToolBridge dataset. To provide further clarity and transparency, we have included a detailed explanation of the transformation process, along with the specific prompts and example input data, in Appendix A.10 of the rebuttal revision.
>
> **Comment2: Evaluate on Tool-Use Benchmarks**
>
> We would like to clarify that the construction of ToolBridge is specifically tailored to support the **tool former** paradigm, which is distinct from the **function calling** paradigm. These paradigms differ fundamentally in their objectives and mechanisms.
>
> - The **tool former** paradigm focuses on enabling large language models (LLMs) to autonomously decide which tools to use, when to use them, and how to integrate their outputs into reasoning processes.
> - In contrast, the **function calling** paradigm emphasizes predefined API interactions, which do not require the explicit reasoning capabilities central to the tool former paradigm.
>
> Given these distinctions, benchmarks such as ToolBench and API-Bank, which are designed to evaluate function-calling systems, are not appropriate for assessing the effectiveness of ToolBridge. To prevent potential misunderstandings, we have explicitly clarified this differentiation in the revised submission, particularly in lines 40–49 of the Introduction section and lines 129–161 of the Related Work section.
>
> Moreover, this distinction reinforces our claim that ToolBridge, to the best of our knowledge, represents the first open-source effort aimed at enabling LLMs to autonomously learn how to utilize external tools, specifically within the framework of the tool former paradigm.
>
> **Comment3: Coding-Focused Base Models**
>
> We would like to clarify that the primary objective of ToolBridge lies in supporting the tool former paradigm, which emphasizes dynamic tool invocation rather than code generation. LLMs trained on ToolBridge are designed to handle diverse problem domains and generate textual content across a wide range of topics. In this context, Python code is dynamically generated to invoke external tools as needed, aiding the reasoning process of LLMs.
>
> Given that coding-focused models are typically optimized for the syntactic and semantic intricacies of programming languages, their specialization would not align well with the broader, domain-agnostic objectives of ToolBridge. As such, training coding-focused base models on ToolBridge would likely yield limited benefits compared to general-purpose models like Llama, which are better suited to the diverse and dynamic requirements of the tool-former paradigm.
>
> **Comment4: Why Use WebQA and FACT for Evaluation?**
>
> The use of WebQA and FACT for evaluation is intended to test the factual retrieval capabilities of LLMs trained on ToolBridge. Specifically, during model inference, LLMs trained on ToolBridge dynamically generate links based on the given context and utilize external tools, such as Python's requests module, to access these links and retrieve factual information. This process enables the model to incorporate externally verified information into its outputs, thereby enhancing the factual accuracy of its results.
>
> **Comment5: Prompt Design and Tool Use Support**
>
> ToolBridge follows the tool-former paradigm, enabling models to autonomously decide whether to use a tool, which tool to invoke, and when to do so, without predefined tools during evaluation. This aligns with the paradigm's emphasis on dynamic, context-driven tool usage.
>
> Regarding the prompts employed during evaluation, for GSM8K, GSM Plus, and MathBench, we adopted a fixed CoT n-shot prompt template as described in the GSM Plus paper. For all other datasets, the models were provided with the question as input, without relying on predefined prompt structures. Details are explicitly outlined in lines 384–391 of the rebuttal revision for clarity.

---

> ### Author Response · Authors · 2024-11-26
> **Response to Reviewer X8CW - Part 2**
>
> **Comment6: Potential Bias Issues**
>
> We would like to clarify that the original data in ToolBridge is sourced from community-contributed datasets and not generated by GPT-4o. GPT-4o’s role is strictly limited to inserting tool calls into the original data at locations where external tools can enhance the reasoning process of LLMs. Importantly, this process is performed on existing data and does not alter the underlying content or its distribution, mitigating concerns about systematic biases being introduced.
>
> To further address the concern, we conducted additional experiments detailed in Appendix A.11 of the rebuttal revision. Specifically, we used Google Gemini, an alternative to GPT-4o, to independently generate three batches of the FACT dataset (referred to as Gemini-FACT-B1, B2, and B3). These datasets are entirely independent of GPT-4o and ToolBridge, serving as an additional check for bias. The evaluation results on these datasets are summarized below:
>
> |Models|SFT data|Gemini-FACT-B1|Gemini-FACT-B2|Gemini-FACT-B3|
> |-|-|-|-|-|
> |Llama3-8B|-|75.8|52.5|60.3|
> |Llama3-8B-Lora|ToolBridge§|83.4|61.7|66.2|
> |Llama3-8B-Lora|ToolBridge|89.2|63.3|71.2|
> |Mistral-7B|-|77.5|59.2|67.8|
> |Mistral-7B-Lora|ToolBridge§|85.8|61.5|70.4|
> |Mistral-7B-Lora|ToolBridge|90.8|64.7|74.7|
>
> The results indicate that models SFT with ToolBridge data consistently outperform their non-ToolBridge counterparts on datasets prepared independently of GPT-4o. This provides strong evidence that the observed improvements are not tied to potential biases introduced by GPT-4o and are instead attributable to the effectiveness of ToolBridge.
>
> **Comment7: Limited accuracy gaps between ToolBridge and ToolBridge$^§$ on standard benchmarks**
>
> We thank the reviewer for highlighting the small performance differences between ToolBridge and ToolBridge$^§$ on standard benchmarks. We would like to address this concern by clarifying the fundamental reasons behind these results and situating them within the broader context of our work.
>
> - **Task-Specific Focus of ToolBridge**. The primary objective of ToolBridge is to enable LLMs to effectively leverage external tools to augment their reasoning processes. This focus diverges from that of many existing standard benchmarks, such as GSM8K, which prioritize evaluating a model’s intrinsic reasoning abilities, particularly in multi-step computation tasks. In these cases, tool usage often contributes only marginal gains, as the benchmarks predominantly assess a model’s reasoning aptitude rather than its tool-integration capabilities. Consequently, the observed limited performance gaps align with our expectations and underscore the distinct objectives of our work and those of standard benchmarks.
>
> - **Absence of Standardized Tool-Usage Evaluations**. A significant limitation in the current landscape of LLMs evaluation is the lack of established benchmarks specifically designed to measure tool-usage capabilities (in the domain of tool former). For instance, even state-of-the-art models like Llama 3.1 are assessed on tool integration primarily through zero-shot evaluations on public datasets (sec. 5.2.7). This underscores the nascent nature of tool-usage evaluation as a research direction and highlights the need for tailored evaluation frameworks.
>
> - **Benefits of Tool Use in Multi-Step Computation**. While the performance gains on existing benchmarks may appear small, enabling tool use can still provide advantages in tasks requiring multi-step computation. For instance, external tools can assist LLMs in handling intricate calculations or processing extensive data.
>
> In conclusion, the limited performance gaps on standard benchmarks should be interpreted in the context of ToolBridge’s broader objective: pioneering the integration of external tools to enhance LLM reasoning. We view this work as an important step toward addressing the open challenge of systematically teaching LLMs to utilize tools effectively.

---

> > ### Comment · Reviewer_X8CW · 2024-11-30
> > **Response to author rebuttal**
> >
> > Thanks for providing the extra results and revising the manuscript to clarify the motivation of the paper on tool former. I increased the presentation score, but I feel there are still weaknesses in the work, which makes me feel it is still marginally below the publishing bar of  ICLR. In particular:
> > 1. Limitation of the benchmark.
> >     1. While I agree on the difference between pure function calling/semantic parsing and the studied tool former approach. I don't think there is a clear distinction between tool use and tool former. Many of the tool use works, including Toolbench [1], gaia[2],and Toolcomp [3], also require dynamically invoking tools, reasoning and planning the tool use, and incorporating the results into the final answer. Although sometimes they are formatted into multiple turns but without additional user input in the middle, I feel they are close to the tool use during the inference loop algorithm described in the paper. Nevertheless, these are definitely related capabilities and seem to be a better fit than the benchmarks (math and webqa) used in the manuscript.
> >     2. For the new dataset, RandomQA seems to be mostly generating basic Python functions, e.g., sort an array, it does not seem to be very aligned with the capability aims to evaluate.
> > 2. Applicability on web data. I agree that there is potential to apply the pipeline to general web data. However, it is still future work and not part of this one. Thus, the limitation of relying on existing SFT data is still there, as there is no experiment backing the hypnosis yet.
> > 3. Baselines. For a fair comparison with the base models, I feel at least some in-context example is needed rather than only presenting a zero-shot experiment. After all, the fine-tuned model is specifically tuned to use tools and format. The LLMs should already have basic tool use/coding ability if prompted to output the prompt, so I feel it is necessary to understand the true gap in call and use tool results, providing decent in-context examples to let the model understand the task.
> >
> > With all these, I appreciate the revision and additional results, but I keep my recommendation as a weak reject.
> >
> > [1] Qin, Yujia, Shihao Liang, Yining Ye, Kunlun Zhu, Lan Yan, Yaxi Lu, Yankai Lin et al. "ToolLLM: Facilitating Large Language Models to Master 16000+ Real-world APIs." In The Twelfth International Conference on Learning Representations.
> > [2] Mialon, Grégoire, Clémentine Fourrier, Thomas Wolf, Yann LeCun, and Thomas Scialom. "GAIA: a benchmark for General AI Assistants." In The Twelfth International Conference on Learning Representations.
> > [3] ToolComp, https://scale.com/leaderboard/tool_use

---

> ### Author Response · Authors · 2024-12-01
> **Response to Reviewer X8CW**
>
> We appreciate the reviewer's response; here is our rebuttal to your reply.
>
> **Comment1: Limitation of the benchmark**
>
> - Differences between "tool use" and "tool former"
>     - In function calling methods like Toolllm, the reasoning process mainly involves selecting the right tool and setting the correct parameters. In contrast, Toolformer not only calls the tool but also reasons about the tool’s output to guide further text generation. This two-step reasoning—first for selecting and calling the tool, then for using the tool's results to generate text—distinguishes Toolformer from methods focused solely on function calling.
>     - Scope of Tool Integration: Tool use systems (e.g., Toolbench, GAIA) rely on predefined, externally structured workflows for tool invocation. Toolformer integrates tool use directly into the model's inference loop, allowing dynamic decisions without external scripting or multi-turn structures.
>     - Tool Selection Granularity: Tool use methods often follow predefined logic for selecting tools based on task-specific needs (e.g., ToolComp). In contrast, Toolformer treats tool invocation as part of language modeling, enabling flexible, context-driven decisions across tasks. BTW, tool use methods rely heavily on strong priors and predefined structures, so it’s unsurprising that they achieve high performance on standard datasets.
>     - Learning vs. Orchestration: Toolformer learns tool use end-to-end, including when and how to invoke tools within generation. Tool use systems rely more on external orchestration, separating reasoning and tool invocation into distinct phases.
>
> - RandomQA: We acknowledge that RandomQA includes some simple questions; however, even these simple questions are challenging for the base model to answer correctly. This further highlights the significance of our open-sourced ToolBridge dataset.
>
> **Comment2: Applicability on web data**
>
> First, in the context of evaluating an open-source dataset, determining whether its data source originates from web text data should not be considered a primary criterion for assessing its contribution, nor is it a particularly significant factor in measuring its value. Currently, in the field of large language models (LLMs), a significant number of open-source datasets are built upon existing open-source datasets, e.g., "Math-LLaVA: Bootstrapping Mathematical Reasoning for Multimodal Large Language Models", "MultiLegalPile: A 689GB Multilingual Legal Corpus", "IndicLLMSuite: A Blueprint for Creating Pre-training and Fine-Tuning Datasets for Indian Languages", "MAmmoTH: Building Math Generalist Models through Hybrid Instruction Tuning", “OpenMathInstruct-1: A 1.8 Million Math Instruction Tuning Dataset”, to name a few.
>
> Secondly, the primary contribution of this work is to address a significant limitation in the existing literature: while prior LLMs (e.g., Llama3.1, Qwen math 2.5, GPT-4o) capable of tool use (in the tool former field) have demonstrated impressive capabilities, none have publicly released a comprehensive data construction pipeline or the associated datasets. To fill this critical gap, our study proposes a systematic methodology for constructing tool former datasets and makes the resulting dataset publicly available, thereby providing a valuable resource for advancing research in this domain.
>
> Finally, utilizing existing SFT data is not a cost-free endeavor. It requires an initial investigation of the data, manual quality evaluation, and subsequent processing and integration before we can proceed to construct our ToolBridge dataset.
>
> **Comment3: Baselines**
>
> First, the experiments in this paper are not limited to zero-shot scenarios; they also include few-shot settings, as shown in Table 6.
>
> Second, you can indeed construct prompts tailored to specific data, predefine certain text formats for tool invocation, and use LLMs to generate these formats to utilize the tools. However, invoking tools in this manner is essentially meaningless, as the results generated by the tool invocation do not provide positive feedback to the subsequent text generation process. Moreover, the reviewer might consider trying to use a base model like LLama3-8B to directly generate text in specific tool invocation formats. The generated content is typically unusable and demonstrates almost no generalization capability.
>
> Finally, we would like to point out that the evaluation approach mentioned by the reviewer is more commonly used in the tool use domain rather than in the tool former paradigm. This is because tool former does not predefine which tools can be used, making such a comparison impractical.

---

### Meta-Review · Area_Chair_rbR7 · 2024-12-16

**Metareview:**

This work creates a tool use dataset by consolidating various non-too-use datasets and applying a systematic conversion process. Experiments on standard math, QA benchmarks, and custom benchmarks reveal consistent performance improvements after fine-tuning with ToolBridge.
The reviewers agree that the dataset coming from this work provides value, but all raised concern on how does this work distinguish from broader tool use scenarios and code generation. Particularly, the work lacks the experiments to demonstrate the benefits of ToolBridge on tool use benchmarks like GAIA, ToolBench and ToolCamp. As reviewer X8CW and NXYV pointed out, there is no clear distinction between tool use/function calling and ToolFormer-like approach.

**Additional Comments On Reviewer Discussion:**

NA

---

### Decision · Program_Chairs · 2025-01-22

Reject